# WARFARE: BREAKING THE WATERMARK PROTECTION OF AI-GENERATED CONTENT

## ABSTRACT

AI-Generated Content (AIGC) is gaining great popularity, with many emerging commercial services using advanced generative models to create realistic images and fluent text. Regulating such content is crucial to prevent policy violations, such as unauthorized commercialization or unsafe content distribution. Watermarking is a promising solution for content attribution and verification, and numerous watermarking approaches have been proposed recently. However, we demonstrate its vulnerability to two key attacks: (1) Watermark removal: the adversary can easily erase the embedded watermark from the generated content and then use it freely bypassing the regulation of the service provider. (2) Watermark forging: the adversary can create illegal content with forged watermarks from another user, causing the service provider to make wrong attributions. We propose `Warfare`, a unified attack framework leveraging a pre-trained diffusion model for content processing and a generative adversarial network for watermark manipulation. Evaluations across datasets and embedding setups show that `Warfare` can achieve high success rates while maintaining the quality of the generated content. We further introduce `Warfare-Plus`, which enhances efficiency without compromising effectiveness.

## 1 INTRODUCTION

Benefiting from the advance of generative deep learning models (Rombach et al., 2022; Touvron et al., 2023), AI-Generated Content (AIGC) has become increasingly prominent. Many commercial services have been released, which leverage large models (e.g., ChatGPT (cha), Midjourney (Mid)) to generate creative content based on users' demands. The rise of AIGC also leads to some legal considerations, and the service provider needs to set up some policies to regulate the usage of generated content. *First*, the generated content is one important intellectual property of the service provider. Many services do not allow users to make it into commercial use (Touvron et al., 2023; Mid). Selling the generated content for financial profit (Sel) will violate this policy and cause legal issues. *Second*, generative models have the potential of outputting unsafe content (Wei et al., 2023; Qi et al., 2023; Liu et al., 2023a; Le et al., 2023), such as fake news (Guo et al., 2021), malicious AI-powered images (Salman et al., 2023; Le et al., 2023), phishing campaigns (Hazell, 2023), and cyberattack payloads (Charan et al., 2023). New laws are established to regulate the generation and distribution of content from deep learning models on the Internet (Gov; Sin; Gui).

As protecting and regulating AIGC become urgent, Google hosted a workshop in June 2023 to discuss the possible solutions against malicious usage of generative models (Barrett et al., 2023). Not surprisingly, the *watermarking* technology is mentioned as a promising defense. By adding invisible specific watermark messages to the generated content (Fernandez et al., 2023; Kirchenbauer et al., 2023; Liu et al., 2023b), the service provider is able to identify the misuse of AIGC and track the corresponding users. A variety of robust watermarking methodologies have been designed, which can be classified into two categories. (1) A general strategy is to make the generative model learn a specific data distribution, which can be decoded by another deep learning model to obtain a secret message as the watermark (Fernandez et al., 2023; Liu et al., 2023b; Zhao et al., 2023b). (2) The service provider can concatenate a watermark embedding model (Zhu et al., 2018; Tancik et al., 2020) after the generative model to make the final output contain watermarks. A very recent work from DeepMind, SynthID Beta (Syn), detects AI-generated images by adding watermarks to the

generated images[1]. According to its description, this service possibly follows a similar strategy as StegaStamp (Tancik et al., 2020), which adopts an encoder to embed watermarks into images and a decoder to identify the embedded watermarks in the given images.

The Google workshop (Barrett et al., 2023) reached the consensus that "existing watermarking algorithms only withstand attacks when the adversary has no access to the detection algorithm", and embedding a watermark to a clean image or text "seems harder for the attacker, especially if the watermarking process involves a secret key". However, we argue that it is not the case. We find that it is easy for an adversary without any prior knowledge to **remove** or **forge** the embedded secret watermark in AIGC, which will break the IP protection and content regulation. Specifically, (1) a watermark removal attack makes the service providers fail to detect the watermarks which are embedded into the AIGC previously, so the malicious user can circumvent the policy regulation and abuse the content for any purpose. (2) A watermark forging attack can intentionally embed the watermark of a different user into the unsafe content without the knowledge of the secret key. This could lead to wrong attributions and frame up that benign user.

Researchers have proposed several methods to achieve watermark removal attacks (Ulyanov et al., 2018; Liang et al., 2021; Li, 2023; Zhao et al., 2023a; Nam et al., 2021; Wang et al., 2022). However, they suffer from several limitations. For instance, some attacks require the knowledge of clean data (Ulyanov et al., 2018; Liang et al., 2021) or details of watermarking schemes (Nam et al., 2021; Wang et al., 2022), which are not realistic in practice. Some attacks take extremely long time to remove the watermark from one image (Li, 2023; Zhao et al., 2023a). Besides, there are currently no studies towards watermark forging attacks. More detailed analysis can be found in Section 2.2.

To remedy the above issues, we introduce `Warfare`, a novel and efficient methodology to achieve both watermark forge and removal attacks against AIGC in a unified manner. The key idea is to leverage a pre-trained diffusion model and train a generative adversarial network (GAN) for erasing or embedding watermarks to AIGC. Specifically, the adversary only needs to collect the watermarked AIGC from the target service or a specific user, without any clean content. Then he can adopt a public diffusion model, such as DDPM (Ho et al., 2020), to denoise the collected data. The preprocessing operation of the diffusion model can make the embedded message unrecoverable from the denoised data. Finally, the adversary trains a GAN model to map the data distribution from collected data to denoised data (for watermark removal) or from denoised data to collected data (for watermark forge). After this model is trained, the adversary can adopt the generator to remove or forge the specific watermark for AIGC. To reduce the time cost and break robust watermarking schemes, which could be resistant against diffusion denoising, we propose `Warfare-Plus` by replacing the preprocessing operation in `Warfare` with a naive unconditional sampling processing.

We evaluate our proposed methods on various datasets (e.g., CIFAR-10, CelebA), and settings (e.g., different watermark lengths, few-shot learning), to show its generalizability. Our results prove that the adversary can successfully remove or forge a specific watermark in the AIGC and keep the content indistinguishable from the original one. This provides concrete evidence that existing watermarking schemes are not reliable, and the community needs to explore more robust watermarking methods. Overall, our contribution can be summarized:

- To the best of our knowledge, it is one of the earliest unified frameworks focusing on both **removing and forging watermarks** in AIGC under a black-box threat model. `Warfare` and `Warfare-Plus` are unified methodologies, which can holistically achieve both attack goals. We disclose the unreliability and fragility of existing watermarking schemes.
- Different from prior attacks, `Warfare` and `Warfare-Plus` **do not require the adversary to have corresponding unwatermarked data or any information about the watermarking schemes**, which is more practical in real-world applications.
- Comprehensive evaluation proves that `Warfare` and `Warfare-Plus` can efficiently remove or forge the watermarks without harming the data quality. The total time cost is analyzed in Appendix E.
- Our methods are effective in the few-shot setting, i.e., it can be **freely adapted to unseen watermarks and out-of-distribution images**. It remains effective for different watermark lengths.

---

[1]Up to the date of writing, SynthID Beta is still a beta product only provided to a small group of users. Since we do not have access to it, we do not include evaluation results with respect to it in our experiments.

## 2 RELATED WORKS

### 2.1 CONTENT WATERMARK

The rapid progress of large and multimodal models has renewed interest in generative systems—e.g., ChatGPT (cha) and Stable Diffusion (Rombach et al., 2022)—capable of producing high-quality images, text, audio, and video (Ho et al., 2020; Touvron et al., 2023; Kong et al., 2021; Ho et al., 2022). Because such AI-Generated Content (AIGC) often carries sensitive and high-value IP, protecting it on public platforms like Twitter and Instagram is crucial (Twi; Ins). A common solution is watermarking, which embeds a secret, unique message for later ownership verification and attribution. Methods fall into (1) post hoc approaches—either visible marks (adding characters/graphics) (Liu et al., 2021; Cheng et al., 2018; Wen et al., 2023) or invisible marks via steganography/signal transforms (Zhu et al., 2018; Tancik et al., 2020; Nam et al., 2021)—and (2) prior approaches where the generator learns to emit decodable, watermarked outputs (Fei et al., 2022; Fernandez et al., 2023; Cui et al., 2023; Zhao et al., 2023b). For GANs, Fei et al. supervise the generator with a watermark decoder (Fei et al., 2022); for diffusion models such as Stable Diffusion, schemes embed predefined bit strings retrievable by a secret decoder (Rombach et al., 2022; Fernandez et al., 2023; Cui et al., 2023; Zhao et al., 2023b). These techniques enable service providers to recognize AIGC from their generative models and, when needed, attribute it to specific user accounts.

### 2.2 WATERMARK ATTACKS

An et al. (2024) introduce a surrogate decoder attack: an adversary trains a substitute watermark detector on watermarked and non-watermarked images, then uses PGD to craft perturbations on the surrogate that transfer to the target detector and flip its decisions. Yang et al. (2024) study an averaging attack, which aggregates many watermarked images, and contrasted with clean counterparts to estimate a fixed watermark pattern that is then subtracted to remove the watermark or added to forge one. While these attacks can be effective against fragile schemes, they falter against robust watermarks. A concurrent work, Soucek et al. (2025), can also be applied to both watermark removal and forgery, but its effectiveness and the resulting image quality are limited. Wang et al. (2021) consider the watermark forging attack. However, they assume the adversary knows the watermarking schemes, which is unrealistic. And they only evaluate LSB- and DCT-based watermarks instead of advanced deep-learning schemes. Müller et al. (2025) propose a black-box forgery attack on semantic watermarks for diffusion models. It perturbs the generation conditions (e.g., prompts or latent codes) to induce the desired watermark. But it's a tailored design to semantic watermarks and failed to remove non-semantic watermarks. Dong et al. (2025) propose WMCopier, which learns to copy the watermark pattern from source images and transfer it to target images. Other prior works mainly focus on the watermark removal attack. These attack solutions can be summarized into three main categories, i.e., image inpainting methods (Ulyanov et al., 2018; Liang et al., 2021) for visible watermarks, denoising methods (Li, 2023; Zhao et al., 2023a), and disrupting methods (Nam et al., 2021; Wang et al., 2022) for invisible watermarks. However, they have several critical drawbacks in practice. Specifically, the image inpainting methods (Ulyanov et al., 2018; Liang et al., 2021) require clean images and watermarked images to train the inpainting model, which is not feasible in the real world, because the user can only obtain watermarked images from the service providers (Mid). Disrupting methods (Nam et al., 2021; Wang et al., 2022) require the user to know the details of the watermarking schemes, which is also difficult to achieve. The most promising method is based on denoising models. For instance, Li (2023) adopted guided diffusion models to purify the watermarked images and minimize the differences between the watermarked images and diffusion model's outputs, and CtrlRegen (Liu et al., 2025) regenerates images from clean noise with controllable guidance to remove the watermark. However, using diffusion models to remove the watermark will require a significant amount of time and often degrade image quality. Our method aims to address these limitations under a black-box threat model.

## 3 PRELIMINARY

### 3.1 SCOPE

In this paper, we target both post hoc and prior watermarking methods. For post hoc methods, we do not consider visible watermarks as they can significantly decrease the visual quality of AIGC, making them less popular for practical adoption. For instance, the Tree-Ring watermark (Wen et al.,

Figure 1: Overview of `Warfare`. (1) Collecting watermarked data from the target AIGC service or Internet. (2) Using a public pre-trained denoising model to purify the watermarked data. (3) Adopting the watermarked and mediator data to train a GAN, which can be used to remove or forge the watermark. $x'$ is the watermarked image. $\hat{x}$ is the mediator image. The subscript $i$ is omitted.

2023) is proven to significantly change both pixel and latent spaces, which is treated as "a visible watermark" (Zhao et al., 2023a). Hence, it is beyond the scope of this paper. For invisible watermarks, we only consider the steganography approach, as it is much more robust and harder to attack than the signal transformation approach (Nam et al., 2021; Wang et al., 2022; Zhao et al., 2023a). We consider watermarks embedded in the generated images. Watermarks in other domains, e.g., language, audio, will be our future work.

## 3.2 Watermark Verification Scheme

We consider the most popular type of secret message used in watermark: bit strings (Fei et al., 2022; Fernandez et al., 2023; Cui et al., 2023; Zhao et al., 2023b). When a service provider $P$ employs a generative model $\mathcal{M}_G$ to generate creative images for public users, $P$ employs a watermarking scheme (e.g., (Fernandez et al., 2023; Liu et al., 2023b)) to embed a secret user-specific bit string $m$ of length $L$ in each image. To verify whether a suspicious image $x^s$ is watermarked by $P$ for a specific user, $P$ uses a decoder $\mathcal{M}_D$ to extract the bit string $m^s$ from $x^s$. Then, $P$ calculates the Hamming Distance between $m$ and $m^s$: $\text{HD}(m, m^s)$. If $\text{HD}(m, m^s) \leq (1 - \tau)L$, where $\tau$ is a pre-defined threshold, $P$ will believe that $x^s$ contains the secret watermark $m$.

## 3.3 Threat Model

**Attack Goals.** A malicious user can break this watermarking scheme with two distinct goals. (1) *Watermark removal attack*: the adversary receives a generated image from the service provider, which contains the secret watermark associated with him. He aims to erase the watermark from the generated image, and then use it freely without the constraint of the service policy, as the provider is not able to identify the watermarks and track him anymore. (2) *Watermark forging attack*: the adversary tries to frame up a victim user by forging the victim's watermark on a malicious image (from another model or created by humans). Then the adversary can distribute the image on the Internet. The service provider will attribute to the wrong user.

**Adversary's capability.** We consider the black-box scenario, where the adversary can only obtain the generated image and has no knowledge of the generative model or watermark scheme. This is practical, as many service providers only release APIs for users without leaking any information about the details of the backend models $\mathcal{M}_G$ and $\mathcal{M}_D$. We further assume that all the generated images from the target service are watermark-protected, so the adversary cannot collect any clean images from the same generative model. These assumptions increase the attack difficulty compared to prior works (Ulyanov et al., 2018; Liang et al., 2021; Nam et al., 2021; Wang et al., 2022).

## 4 Warfare: A Unified Attack Methodology

We introduce `Warfare` to manipulate watermarks with the above goals. Let $x_i$ denote a clean image, and $x'_i$ denote the corresponding watermarked image. These two images are visually indistinguishable. Our goal is to establish a bi-directional mapping $x_i \longleftrightarrow x'_i$. For the watermark removal attack, we can derive $x_i$ from $x'_i$. For the watermark forging attack, we can construct $x'_i$ from $x_i$.

However, it is challenging for the adversary to identify the relationship between $x_i$ and $x'_i$, as he has no access to the clean image $x_i$. To address this issue, the adversary can adopt a pre-trained denoising model to convert $x'_i$ into a mediator image $\hat{x}_i$. Due to the denoising operation, $\hat{x}_i$ is visually different from $x_i$, but does not contain the watermark. It will follow a similar "non-watermarked" distribution as $x_i$. Then he can train a GAN between $x_i$ and $x'_i$, which is guided by $\hat{x}_i$. Figure 1 shows the overview of `Warfare`, consisting of three steps. Below, we describe the details.

Table 1: `Warfare` under the different number of collected images on CIFAR-10. The length of embedded bits is 8.

| # of Samples | Original | | | | | Watermark Remove | | | | | Watermark Forge | | | | |
|---|---|---|---|---|---|---|---|---|---|---|---|---|---|---|---|
| (bit length = 8bit) | Bit Acc | FID | PSNR | SSIM | CLIP | Bit Acc | FID↓ | PSNR↑ | SSIM↑ | CLIP↑ | Bit Acc↑ | FID↓ | PSNR↑ | SSIM↑ | CLIP↑ |
| 5000 | | | | | | 49.42% | 20.75 | 24.64 | 0.83 | 0.92 | 96.11% | 18.86 | 24.36 | 0.83 | 0.93 |
| 10000 | | | | | | 50.68% | 23.76 | 24.31 | 0.82 | 0.90 | 98.63% | 15.68 | 24.70 | 0.81 | 0.94 |
| 15000 | 100.00% | 6.19 | 25.23 | 0.83 | 0.99 | 59.88% | 20.32 | 22.87 | 0.80 | 0.92 | 97.80% | 25.34 | 24.55 | 0.80 | 0.92 |
| 20000 | | | | | | 54.59% | 22.90 | 24.93 | 0.84 | 0.90 | 95.99% | 23.56 | 23.74 | 0.80 | 0.92 |
| 25000 | | | | | | 47.80% | 18.42 | 23.59 | 0.83 | 0.91 | 97.84% | 21.09 | 24.94 | 0.82 | 0.93 |

Table 2: Performance of `Warfare` under different bit lengths on CIFAR-10. The number of images for the adversary is 25,000. ↓ means lower is better. ↑ means higher is better.

| Bit Length | Original | | | | | Watermark Remove | | | | | Watermark Forge | | | | |
|---|---|---|---|---|---|---|---|---|---|---|---|---|---|---|---|
| | Bit Acc | FID | PSNR | SSIM | CLIP | Bit Acc | FID↓ | PSNR↑ | SSIM↑ | CLIP↑ | Bit Acc ↑ | FID↓ | PSNR↑ | SSIM↑ | CLIP↑ |
| 4 bit | 100.00% | 4.22 | 27.81 | 0.89 | 0.99 | 52.53% | 16.36 | 24.51 | 0.86 | 0.92 | 95.76% | 17.59 | 26.70 | 0.88 | 0.94 |
| 8 bit | 100.00% | 6.19 | 25.23 | 0.83 | 0.99 | 47.80% | 18.42 | 23.59 | 0.83 | 0.91 | 97.84% | 21.09 | 24.94 | 0.82 | 0.93 |
| 16 bit | 100.00% | 11.34 | 22.71 | 0.73 | 0.98 | 50.10% | 24.63 | 23.44 | 0.77 | 0.91 | 92.23% | 18.34 | 25.84 | 0.83 | 0.94 |
| 32 bit | 99.99% | 28.76 | 19.99 | 0.53 | 0.96 | 53.64% | 25.33 | 21.17 | 0.64 | 0.91 | 90.14% | 31.13 | 23.41 | 0.71 | 0.93 |

## 4.1 DATA COLLECTION

The adversary collects a set of images $x_i'$ generated by the target service provider for one user. All the collected data contain one specific watermark $m$ associated with this user. For the watermark removal attack, the adversary can query the service to collect the watermarked images with his own account, from which he aims to remove the watermark. For the watermark forging attack, the adversary can possibly collect such data from the victim user's social account. This is feasible as people enjoy sharing their created content on the Internet and adding tags to indicate the used service[2]. Then the adversary can forge the watermark of the victim user on any images to cause wrong attribution. In either case, a dataset $\mathcal{X}' = \{x_i'|x_i' \sim (\mathcal{M}_G, m)\}$ is established, where $\mathcal{M}_G$ is the service provider's generative model.

## 4.2 DATA PRE-PROCESSING

Given the collected watermarked dataset $\mathcal{X}'$, since the adversary does not have the corresponding non-watermarked dataset $\mathcal{X}$, he cannot directly build the mapping. Instead, he can adopt a public pre-trained denoising model $\mathcal{H}$ to preprocess $\mathcal{X}'$ and obtain the corresponding mediator dataset $\hat{\mathcal{X}}$. The goal of the denoising model is to remove the watermark $m$ from $\mathcal{X}'$. Since existing watermarking schemes are designed to be very robust, we have to increase the denoising strength significantly, in order to distort the embedded watermark. Therefore, we first add very large-scale noise $\epsilon_i$ into $x_i'$ and then apply a diffusion model $\mathcal{H}$ to denoise the images, i.e., $\hat{\mathcal{X}} = \{\hat{x}_i|\mathcal{H}(x_i' + \epsilon_i) = \hat{x}_i, x_i' \in \mathcal{X}', \epsilon_i \in \mathcal{N}(\mathbf{0}, \mathbf{I})\}$. This will make $\hat{x}_i$ highly visually different from $x_i'$ and $x_i$. Figure 6 shows some visualization results of $x_i'$ and $\hat{x}_i$, and we can observe that they keep some similar semantic information but look very different. Table 3 proves that $\hat{x}_i$ does not contain any watermark information due to the injected large noise and strong denoising operation.

The mediator dataset $\hat{\mathcal{X}}$ can be seen as being drawn from the same "non-watermarked" distribution as $\mathcal{X}$, which is different from $\mathcal{X}'$ of the "watermarked" distribution. Therefore, it can help discriminate watermarking images from non-watermarked images and build connections between them. We provide a theoretical analysis supporting this claim in Appendix K.

## 4.3 MODEL TRAINING

With the watermarked data $x'$ and non-watermarked data $\hat{x}$, the adversary can train a GAN model to add or remove watermarks. This GAN model consists of a generator $\mathcal{G}$ and a discriminator $\mathcal{D}$: $\mathcal{G}$ is used to generate $x$ from $x'$ (watermark removal) or generate $x'$ from $x$ (watermark forging); $\mathcal{D}$ is used to discriminate whether the input is drawn from the distribution of watermarked images $x'$ or the distribution of non-watermarked images $\hat{x}$. Below, we describe these two attacks.

**Watermark removal attack**. In this attack, the generator $\mathcal{G}$ is built to obtain $x$ from $x'$, i.e., $x = \mathcal{G}(x')$, where $x'$ and $x$ should be visually indistinguishable. $x$ generated by $\mathcal{G}$ should make $\mathcal{D}$ believe it is from the same non-watermarked image distribution as $\hat{x}$, because $x$ should be a non-watermarked image. Meanwhile, $\mathcal{D}$ should recognize $x$ as a watermarked image, since it is very

---

[2]The adversary can collect watermarked content with his own account as well because our method shows strong few-shot power, which can be found in our experiments. The adversary can adopt very few samples to fit an unseen watermark.

Table 3: Results on CelebA. The bit string length is 32 bits. Best results in **Bold**. Second best results with Underline.

| Methods | Original | | | | | Watermark Remove | | | | | Watermark Forge | | | | |
|---|---|---|---|---|---|---|---|---|---|---|---|---|---|---|---|
| | Bit Acc | FID | PSNR | SSIM | CLIP | Bit Acc | FID | PSNR | SSIM | CLIP | Bit Acc | FID | PSNR | SSIM | CLIP |
| CenterCrop | | | | | | 59.89% | - | - | - | 0.90 | 48.33% | - | - | - | 0.93 |
| GaussianNoise | | | | | | 99.92% | 53.80 | 24.97 | 0.71 | 0.86 | 52.28% | 47.07 | 28.64 | 0.75 | 0.89 |
| GaussianBlur | | | | | | 100.00% | 25.09 | 26.26 | 0.84 | 0.86 | 52.10% | 21.18 | 28.17 | 0.88 | 0.89 |
| JPEG | | | | | | 99.27% | 17.42 | 28.40 | 0.89 | 0.89 | 52.19% | 9.96 | 33.36 | 0.94 | 0.90 |
| Brightness | | | | | | 100.00% | 4.26 | 19.70 | 0.87 | 0.95 | 52.28% | 0.39 | 21.16 | 0.91 | 0.98 |
| Gamma | | | | | | 100.00% | 4.43 | 22.93 | 0.88 | 0.96 | 52.32% | 0.26 | 25.71 | 0.93 | 0.99 |
| Hue | 100.00% | 4.25 | 30.7 | 0.94 | 0.96 | 99.99% | 5.93 | 26.84 | 0.93 | 0.94 | 52.21% | 1.60 | 32.06 | 0.98 | 0.97 |
| Contrast | | | | | | 100.00% | 4.26 | 24.28 | 0.85 | 0.95 | 52.33% | 0.25 | 27.62 | 0.90 | 0.98 |
| $DM_s$ | | | | | | 67.82% | 73.30 | 20.61 | 0.62 | 0.69 | 48.78% | 68.91 | 20.89 | 0.64 | 0.70 |
| $DM_l$ | | | | | | **47.20%** | 82.38 | 15.76 | 0.34 | 0.67 | 45.96% | 79.06 | 15.81 | 0.34 | 0.68 |
| $VAE_{SD}$ | | | | | | 65.32% | 43.21 | 19.57 | 0.66 | 0.76 | 49.36% | 40.50 | 19.84 | 0.68 | 0.77 |
| $VAE_C$ | | | | | | 54.36% | 115.79 | 17.42 | 0.43 | 0.72 | 53.90% | 115.19 | 17.47 | 0.43 | 0.72 |
| Warfare | | | | | | 51.98% | 9.93 | 26.61 | 0.91 | 0.90 | **99.11%** | 8.75 | 24.92 | 0.90 | 0.92 |

close to $x'$. Therefore, the loss functions $L_\mathcal{G}$ for $\mathcal{G}$ and $L_\mathcal{D}$ for $\mathcal{D}$ are:

$$L_\mathcal{D} = -\mathbb{E}_{\hat{x} \in \hat{\mathcal{X}}} \mathcal{D}(\hat{x}) + \mathbb{E}_{x' \in \mathcal{X}'} \mathcal{D}(\mathcal{G}(x'))$$
$$+ w_\mathcal{D} \mathbb{E}_{\hat{x} \in \hat{\mathcal{X}}, x' \in \mathcal{X}'} \nabla_{\alpha x' + (1-\alpha)\hat{x}} \mathcal{D}(\alpha x' + (1-\alpha)\hat{x}),$$
$$L_{\mathcal{G}_x} = \mathbb{E}_{x' \in \mathcal{X}'}[\mathrm{L}_1(\mathcal{G}(x'), x') + \mathrm{MSE}(\mathcal{G}(x'), x')$$
$$+ \mathrm{LPIPS}(\mathcal{G}(x'), x')],$$
$$L_{\mathcal{G}_D} = -w_\mathcal{G} \mathbb{E}_{x' \in \mathcal{X}'} \mathcal{D}(\mathcal{G}(x')), \quad L_\mathcal{G} = L_{\mathcal{G}_D} + w_x L_{\mathcal{G}_x},$$

where $w_\mathcal{D}$, $w_\mathcal{G}$, and $w_x$ are the weights for losses and $\alpha$ is a random variable between 0 and 1 (Arjovsky et al., 2017)[3]. $\mathrm{L}_1$ is the $L_1$-norm, MSE is the mean squared error loss, and LPIPS is the perceptual loss (Zhang et al., 2018). They can guarantee the quality of the generated image $x$.

**Watermark forging attack**. In this attack, the generator $\mathcal{G}$ is built to obtain $\hat{x}'$ from $\hat{x}$, i.e., $\hat{x}' = \mathcal{G}(\hat{x})$, where $\hat{x}'$ and $\hat{x}$ should be visually indistinguishable. $\hat{x}'$ is the watermarked version of $\hat{x}$. $\hat{x}'$ generated by $\mathcal{G}$ should make $\mathcal{D}$ believe it is from the same watermarked image distribution as $x'$, because $\hat{x}'$ should be a watermarked image. But $\mathcal{D}$ should recognize $\hat{x}'$ as a non-watermarked image, since it is very close to $\hat{x}$. The loss functions $L_\mathcal{G}$ for $\mathcal{G}$ and $L_\mathcal{D}$ for $\mathcal{D}$ are:

$$L_\mathcal{D} = -\mathbb{E}_{x' \in \mathcal{X}'} \mathcal{D}(x') + \mathbb{E}_{\hat{x} \in \hat{\mathcal{X}}} \mathcal{D}(\mathcal{G}(\hat{x}))$$
$$+ w_\mathcal{D} \mathbb{E}_{\hat{x} \in \hat{\mathcal{X}}, x' \in \mathcal{X}'} \nabla_{\alpha x' + (1-\alpha)\hat{x}} \mathcal{D}(\alpha x' + (1-\alpha)\hat{x}),$$
$$L_{\mathcal{G}_x} = \mathbb{E}_{\hat{x} \in \hat{\mathcal{X}}}[\mathrm{L}_1(\mathcal{G}(\hat{x}), \hat{x}) + \mathrm{MSE}(\mathcal{G}(\hat{x}), \hat{x}) + \mathrm{LPIPS}(\mathcal{G}(\hat{x}), \hat{x})],$$
$$L_{\mathcal{G}_D} = -w_\mathcal{G} \mathbb{E}_{\hat{x} \in \hat{\mathcal{X}}} \mathcal{D}(\mathcal{G}(\hat{x})), \quad L_\mathcal{G} = L_{\mathcal{G}_D} + w_x L_{\mathcal{G}_x}.$$

The notations are the same as these in the watermark removal attack. It is easy to find that for both types of attacks, the training framework can be seen as a *unified* one, because the adversary only needs to replace $x'$ with $\hat{x}$ or replace $\hat{x}$ with $x'$, to switch to another attack.

### 4.4 WARFARE-PLUS WITH HIGHER TIME EFFICIENCY

In `Warfare`, we adopt a pre-trained diffusion model to purify the watermarked data and obtain the mediator images. This brings additional time cost, which reduces the overall time efficiency of our proposed method. Additionally, advanced watermarking schemes in the future which can defend against diffusion denoising will be resistant to our method as well. To further improve the time efficiency and enhance the attacking effectiveness, we propose `Warfare-Plus` by revising the data pre-processing process. We find that the purified images are not essential in our attack framework. Therefore, we directly adopt an open-sourced Stable Diffusion 1.5 (SD1.5) (sd1) to generate images without conditional prompts as the mediator images. Compared with `Warfare`, we only improve the data pre-processing step and keep other steps unchanged.

## 5 EVALUATIONS

### 5.1 EXPERIMENT SETUP

**Datasets.** We mainly consider two datasets: CIFAR-10 and CelebA (Liu et al., 2015). CIFAR-10 contains 50,000 training images and 10,000 test images with a resolution of 32*32. CelebA is a celebrity faces dataset, which contains 162,770 images for training and 19,867 for testing, resized at a resolution of 64*64 in our experiments. We randomly split the CIFAR-10 training set into two disjoint parts, one of which is to train the service provider's model and another is used by the adversary. Similarly, we randomly pick 100,000 images for the service provider and 10,000 images

---

[3]We slightly modify the discriminator loss for large-resolution images to stabilize the training process. Details are in Appendix A.

for the adversary from the CelebA training set. Furthermore, we also consider a more complex dataset with high resolution (256*256), LSUN (Yu et al., 2015). Furthermore, we also collect some generated images from Stable Diffusion (Rombach et al., 2022) to verify the effectiveness of our method in more complex situations. Details can be found in Appendix G. To enhance the connections with AIGC, we evaluate our method on generative model generated images in Section 5.4 and Appendix C, in which we consider images generated by GANs, conditional diffusion models, and popular Stable Diffusion models.

**Watermarking Schemes.** Considering the watermark's expandability to multiple users, we mainly adopt the post hoc manner, i.e., adding user-specific watermarks to the generated images. We adopt StegaStamp (Tancik et al., 2020), a state-of-the-art and robust method for embedding bit strings into given images, which is proved to be the most effective watermarking embedding method against various removal attacks (Zhao et al., 2023a). **On the other hand, watermarking schemes, such as RivaGAN (Zhang et al., 2019), SSL (Fernandez et al., 2022), and Tree-Ring (Wen et al., 2023) have been shown to be not robust (Zhao et al., 2023a; An et al., 2024). Therefore, we only consider breaking watermarking schemes, which have not been broken before.** Another post hoc scheme is Stable Signature (Fernandez et al., 2023), which is proposed for Stable Diffusion models, specifically. The model owner trains a latent decoder for Stable Diffusion models, which can add a pre-fixed bit string to the generated image. We also provide two case studies to explore the prior manner, which directly generates images with watermarks for our case studies. We follow previous works (Fei et al., 2022; Zhao et al., 2023b) to embed a secret watermark to WGAN-div (Wu et al., 2018) and EDM (Karras et al., 2022).

**Baselines.** To the best of our knowledge, `Warfare` is among the earliest to both remove or forge a watermark in images under a pure black-box threat model. Therefore, we consider some potential baseline attack methods under the same assumptions and attacker's capability, i.e., having only watermarked images. The removal baseline methods can be classified into three groups. (1) Image transformation methods: we consider modifying the properties of the given image, such as resolution, brightness, and contrast. We also consider image compression (e.g., JPEG) and image disruptions (e.g., Gaussian blurring, adding Gaussian noise). (2) Diffusion model methods (Li, 2023): we directly adopt a pre-trained unconditional diffusion model (DiffPure (Nie et al., 2022)) to modify the given image, which does not require to train a diffusion model from scratch and does not need clean images. We also evaluate CtrlRegen (Liu et al., 2025) to regenerate images from clean noise with controllable guidance (3) VAE model methods (Zhao et al., 2023a): we directly adopt two different VAE models. One is from the Stable Diffusion (Rombach et al., 2022), which is named $VAE_{SD}$. Another one is trained on CelebA, which is named $VAE_C$. Specifically, both diffusion models and VAE models are not trained or fine-tuned for watermark removal or forge due to the black-box threat model. We do not adopt guided diffusion models or conditional diffusion models as (Li, 2023) did as well. When attacking Stable Signature, we use a pretrained diffusion model based on ImageNet and the VAE from Stable Diffusion 1.4, as the generated images have larger resolution. When using the diffusion model, we set the noise scale as 75 and set the number of sampling step as 15. The results from pre-trained diffusion models are various on different datasets, which are discussed in Appendix D. We include two forgery baselines Müller et al. (2025) and WMCopier (Dong et al., 2025) under their default settings. We also evaluate two attacks designed to both remove and forge watermarks on the AIGC dataset: the surrogate decoder attack (An et al., 2024), the averaging attack (Yang et al., 2024), and our concurrent work Soucek et al. (2025), all reproduced with the authors' default configurations. For watermark removal, the watermarked images are inputs for the attacks; for watermark forgery, the clean images are inputs for the attacks.

**Implementation.** We adopt DiffPure (Nie et al., 2022) as the diffusion model used in the second step of `Warfare` **without any fine-tuning**. The diffusion model used in DiffPure depends on the domain of watermarked images. For example, if the watermarked images are human faces from CelebA and FFHQ, we use a diffusion model trained on CelebA. For Stable Signature and Tree-Ring, we use a pretrained diffusion model based on ImageNet in `Warfare`, and directly adopt an open-sourced Stable Diffusion 1.5 (SD1.5) (sd1) to generate images without conditional prompts as the mediator images in `Warfare-Plus`. As the adversary does not have any knowledge of the watermarking scheme, it is important to decide which checkpoint should be used in the attack. We provide a simple way to help the adversary select a checkpoint during the training process in Appendix B. More details can be found in Appendix A, including hyperparameters and bit strings.

Table 4: Results of attacking Stable Signature on Stable Diffusion 2.1.

| Method | Origin | | Watermark Remove | | Watermark Forge | |
|---|---|---|---|---|---|---|
| | Bit Acc | FID | Bit Acc | FID | Bit Acc | FID |
| DM | | | 48.29% | 8.77 | - | - |
| VAE | | | 52.69% | 8.72 | - | - |
| CtrlRegen | | | 49.17% | 8.73 | - | - |
| Müller et al. | | | - | - | 47.45% | 13.01 |
| WMCopier | 100% | 7.65 | - | - | 95.23% | 5.72 |
| Averaging | | | 99.95% | 11.48 | 50.10% | 11.95 |
| Surrogate | | | 99.87% | 8.08 | 51.89% | 0.98 |
| Soucek et al. | | | 99.33% | 14.31 | 73.88% | 4.44 |
| Warfare | | | 49.22% | **8.07** | **99.08%** | **0.78** |
| Warfare-Plus | | | **49.95%** | 8.45 | 97.03% | 1.22 |

**Metrics.** To fairly evaluate our proposed Warfare, we consider five metrics to measure its performance from different perspectives. To determine the quality of the watermark removal (forging) task, we adopt **Bit Acc**, which can be calculated as $\text{Bit Acc}(m, m') = \frac{|m| - \text{HD}(m, m')}{|m|} \times 100\%$, where $\text{HD}(\cdot, \cdot)$ is the Hamming Distance. If $\text{Bit Acc}(m, m') \geq \tau$, verification will pass. Otherwise, it will fail. In our experiments, $\tau = 80\%$. To evaluate the quality of the images generated by Warfare and the baselines, we adopt the Fréchet Inception Distance (FID) (Heusel et al., 2017), the peak signal-to-noise ratio (PSNR) (Horé & Ziou, 2010), and the structural similarity index (SSIM) (Horé & Ziou, 2010). Furthermore, we consider the semantic information inside the images, which is evaluated by CLIP (Radford et al., 2021). For the FID, PSNR, SSIM, and CLIP scores, we compute the results between clean images and watermarked images for the watermarking scheme, and between clean images and images after removal or forge attacks. For watermark removal, a lower bit accuracy is better. For watermark forging, a higher bit accuracy is better. For all tasks, a higher PSNR, SSIM, and CLIP score is better. And a lower FID is better.

## 5.2 ABLATION STUDY

In this section, we explore the generalizability of Warfare under the views of the length of the embedding bits and the number of collected images. In Table 2, we show the results of Warfare at different lengths of embedded bits. The results indicate that Warfare is robust for different secret message lengths. Specifically, when the length of the embedded bits increases, Warfare can still achieve good performance on watermark removing or forging and make the transferred images keep high quality and maintain semantic information. In Table 1, we present the results when the adversary uses the different numbers of collected images as his training data. The results indicate that even with limited data, the adversary can remove or forge a specific watermark without harming the image quality, which proves that our method can be a real-world threat. Therefore, our proposed Warfare has outstanding flexibility and generalizability under a practical threat model. We further prove its extraordinary few-shot generalizability for unseen watermarks in Section 5.3.

## 5.3 RESULTS ON POST HOC MANNERS

Here, we focus on post hoc manners, i.e., adding watermarks to AIGC with an embedding model. Because the post hoc watermarking scheme can freely change the embedding watermarks, we evaluate Warfare under few-shot learning to show the capability of adapting to unseen watermarks.

**Results on CelebA.** We consider two different lengths of the embedding bits, i.e., 32-bit and 48-bit. Furthermore, we do not consider the specific coding scheme, including the source coding and the channel coding. Tables 3 and 7 compare Warfare and the baseline methods on the watermark removal task and the watermark forging task, respectively. We notice that the watermark embedding method is robust against various image transformations. Using image transformations cannot simply remove or forge a specific watermark in the given images[4]. For methods using diffusion models, we consider two settings, i.e., adding large noise to the input ($\text{DM}_l$) and adding small noise to the input ($\text{DM}_s$). Especially, we use the same setting as $\text{DM}_l$ in the second step of Warfare to generate images. Although diffusion models can easily remove the watermark from the given images under both settings, the generated images are visually different from the input images, causing a low PSNR, SSIM, and CLIP score. Furthermore, the FID indicates that the diffusion model will cause

---

[4]We omit the results with image transformations in the following tables to save space.

a distribution shift compared to the clean dataset. Nevertheless, we find that $DM_l$ and $DM_s$ can maintain high image quality while successfully removing watermarks on other datasets, which we discuss in Appendix D. The results make us reflect on the generalizability of diffusion models on different datasets and watermarking schemes. However, evaluating all accessible diffusion models on various datasets and watermarking schemes will take months. Therefore, we leave it as future work to deeply study the diffusion models in the watermarking removal task. On the other hand, forging a specific unknown watermark is non-trivial and impossible for both image transformation methods and diffusion models.

Our `Warfare` gives an outstanding performance in both tasks and maintains good image quality as well. However, we notice that as the length of the embedded bit string increases, it becomes more challenging to forge or remove the watermark. That is the reason that under 48-bit length, our `Warfare` has a little performance drop on both tasks with respect to bit accuracy and image quality. We provide visualization results in the following content to prove images generated by `Warfare` are still visually close to the given image under a longer embedding length. More importantly, `Warfare` is time-efficient compared to diffusion model methods. The results are in Appendix E.

**Few-Shot Generalization.** In real-world applications, large companies can assign a unique watermark for every account or change watermarks periodically. Therefore, it is important to study the few-shot power of `Warfare`, i.e., fine-tuning `Warfare` with several new data with an unseen watermark to achieve outstanding watermark removal or forging abilities for the unseen watermark. In our experiments, we mainly consider embedding a 32-bit string into clean images. Then, we fine-tune the model in Table 3 to fit new unseen watermarks. In Table 8, we present the results under 10, 50, and 100 training data for watermark removal and forging. The results indicate that the watermark removal task is much easier than the watermark forging task. Furthermore, with more accessible data, both bit accuracy and image quality can be improved. It is worth noticing that, even with limited data, `Warfare` can successfully remove or forge an unseen watermark and maintain high image quality. The results prove that our proposed method has strong few-shot generalization power to meet practical usage.

Besides, we provide evaluation results on prior watermarking strategies in Appendix C. To better compare the image quality of `Warfare` with other baselines, we show the visualization results in Appendix H. In Appendix I, we discuss the potential defenses to mitigate the attacks. Overall, `Warfare` shows good effectiveness on both types of attack for different watermarking schemes.

## 5.4 WARFARE AND WARFARE-PLUS ON AIGC DATASET

Beyond real-world data distributions, an even more critical area of focus is watermarking AIGC. Compared to the datasets used in our earlier experiments, AI-generated images often feature higher resolutions and more intricate details. Thus, it is essential to evaluate both the effectiveness and efficiency of our method in this context. In this section, we target Stable Signature, a watermarking scheme specifically designed for Stable Diffusion models, and present the results in Table 4. Both the three regeneration removal baselines and our methods successfully remove the watermark. However, `Warfare` and `Warfare-Plus` obtain substantially better FID scores, indicating that our GAN-based framework not only removes the watermark effectively but also preserves image quality more faithfully. Müller et al. (2025) attains only 47.45% forgery bit accuracy, consistent with its design for semantic watermarking. WMCopier is a strong forgery baseline, yet it still lags behind our methods in both accuracy and image fidelity. Neither the averaging attack nor the surrogate-detector attack succeeds in breaking Stable Signature. Soucek et al. (2025) also fails at watermark removal. Although it achieves 73.88% forgery bit accuracy, this is still far from the 100% target, and it produces worse image quality with 4.44 FID compared with our `Warfare` and `Warfare-Plus`.

The findings support our claim that our method is the first to offer a practical, black-box solution for both removal and forgery of AIGC watermarking schemes with high attack success and image quality. Even when training GAN models on high-resolution datasets, we achieve models that perform well in both preserving image quality and removing (or forging) watermarks. However, the time cost increases significantly as image resolution grows. Table 11 provides a detailed breakdown of the computational overhead for each attack step, highlighting that the data preprocessing phase is a bottleneck that impacts the efficiency of `Warfare`. To address this, we propose `Warfare-Plus`, an enhanced version of `Warfare`. While `Warfare-Plus` requires longer training times to en-

sure GAN convergence, it significantly reduces the preprocessing time. Compared to `Warfare`, `Warfare-Plus` decreases the overall time cost—including preprocessing, model training, and inference—by 80% to 85%, while maintaining strong attack performance. For visual confirmation, Appendix H includes sample images to illustrate the image quality. These results confirm that both `Warfare` and `Warfare-Plus` preserve image quality effectively, allowing for the flexible generation of watermarked and non-watermarked images.

## 6 LIMITATIONS AND CONCLUSIONS

In this paper, we consider a practical threat to AIGC protection and regulation schemes, which are based on the state-of-the-art **robust and invisible** watermarking technologies. We introduce `Warfare` and its variant `Warfare-Plus`, a unified attack framework to effectively remove or forge watermarks over AIGC while maintaining good image quality. With our method, the adversary only requires watermarked images without their corresponding clean ones, making it a real-world threat. Through comprehensive experiments, we prove that it has strong few-shot generalization abilities to fit unseen watermarks, which makes it more powerful. Furthermore, we show that it can easily replace a watermark in the collected data with another new one, in Appendix F. We discuss the potential usage of `Warfare` and `Warfare-Plus` for larger-resolution and more complex images, in real-world scenarios. Further improvement over `Warfare` and `Warfare-Plus` is probable with more advanced GAN structures and training strategies.

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

## A  EXPERIMENT SETTINGS

**Model Structures.** For CIFAR-10 and CelebA, we choose different architectures for generators and discriminators to stabilize the training process. Specifically, when training models on CIFAR-10, we use the ResNet-based generator architecture (Zhu et al., 2017) with 6 blocks. As the CelebA images have higher resolution, we use the ResNet-based generator architecture (Zhu et al., 2017) with 9

| Experiment | Watermark Remove | | Watermark Forge | |
|---|---|---|---|---|
| | $w_{\mathcal{G}}$ | $w_x$ | $w_{\mathcal{G}}$ | $w_x$ |
| CIFAR-10 4bit | 500 | 10 | 500 | 5 |
| CIFAR-10 8bit | 800 | 15 | 500 | 10 |
| CIFAR-10 16bit | 500 | 40 | 150 | 40 |
| CIFAR-10 32bit | 100 | 40 | 100 | 40 |
| CIFAR-10 5000 data | 800 | 15 | 500 | 10 |
| CIFAR-10 10000 data | 800 | 15 | 600 | 20 |
| CIFAR-10 15000 data | 500 | 15 | 500 | 10 |
| CIFAR-10 20000 data | 800 | 15 | 500 | 15 |
| CIFAR-10 25000 data | 800 | 15 | 500 | 10 |
| CelebA 32bit | 10 | 120 | 1 | 10 |
| CelebA 48bit | 10 | 200 | 1 | 10 |
| Few-Shot 10 Images | 10 | 200 | 1 | 10 |
| Few-Shot 50 Images | 10 | 200 | 1 | 10 |
| Few-Shot 100 Images | 10 | 200 | 1 | 10 |
| WGAN-div | 10 | 120 | 1 | 10 |
| EDM | 1 | 10 | 100 | 1 |
| Stable Signature (`Warfare`) | 10 | 5 | 10 | 100 |
| Stable Signature (`Warfare-Plus`) | 10 | 10 | 10 | 100 |

Table 5: Hyperparameter settings in our experiments for watermark removal and watermark forging.

blocks. For the discriminators, we use a simple model containing 4 convolutional layers for CIFAR-10. And for CelebA, a simple discriminator cannot promise a stable training process. Therefore, we use a ResNet-18 (He et al., 2016). To improve the quality of generated images, we follow the residual training manner, that is, the output from the generators will be added to the original input.

**Hyperparameters.** We use different hyperparameters for CIFAR-10 and CelebA, respectively. When training models on CIFAR-10, we use RMSprop as the optimizer for both the generator and the discriminator. The learning rate is 0.0001, and the batch size is 32. We set $w_{\mathcal{D}} = 10$, and the total number of training epochs is 1,000. We update the generator's parameters after 5 times of updating of the discriminator's parameters. For CelebA, we adopt Adam as our model optimizer. The learning rate is 0.003, and the batch size is 16. We replace the discriminator loss with the one from StyleGAN (Karras et al., 2019) with $w_{\mathcal{D}} = 5$, and the total number of training epochs is 1,000. We update the generator's parameters after updating the discriminator's parameters. We present $w_{\mathcal{G}}$ and $w_x$ in Table 5 used in our experiments. We choose the best model based on the image quality.

**Baseline Settings.** For image transformation methods, we mainly adopt torchvision to implement attacks. To adjust brightness, contrast, and gamma, the changing range is randomly selected from 0.5 to 1.5. To adjust the hue, the range is randomly selected from -0.1 to 0.1. For center-cropping, we randomly select the resolution from 32 to 64. For the Gaussian blurring, we randomly choose the Gaussian kernel size from 3, 5, and 7. For adding Gaussian noise, we randomly choose $\sigma$ from 0.0 to 0.1. For JPEG compression, we randomly selected the compression ratio from 50 to 100. When evaluating the results of image transformation methods, we run multiple times and use the average results. For diffusion methods $\text{DM}_l$, we set the sample step as 30 and the noise scale as 150. For diffusion methods $\text{DM}_s$, we set the sample step as 200 and the noise scale as 10. Specifically, we use $\text{DM}_l$ in the second step of `Warfare`. Considering using diffusion models to generate images is very time-consuming, we randomly select 1,000 images from the test set to obtain the results for diffusion models.

**Stable Signature.** The diffusion model used in Stable Signature is Stable Diffusion 2.1 (SD2.1) (sd2). During the generation process, we adopt the unconditional generation approach by setting the prompt empty to obtain images with 512*512 resolution. We sample 10,000 watermarked images for our attack method.

**Embedded Bits.** In Table 6, we list the bit strings embedded in the images in our experiments.

| Experiment | Bit String |
|---|---|
| CIFAR-10 4bit | 1000 |
| CIFAR-10 8bit | 10001000 |
| CIFAR-10 16bit | 1000100010001000 |
| CIFAR-10 32bit | 10001000100010001000100010001000 |
| CelebA 32bit | 10001000100010001000100010001000 |
| CelebA 48bit | 100010001000100010001000100010001000100010001000 |
| Few-Shot | 11100011101010101000010000001011 |
| WGAN-div | 10001000100010001000100010001000 |
| EDM | 0100010001000010111010111111110011101000001111101101010110000000 |
| Stable Signature | 111010110101000001010111010011010100010000100111 |

Table 6: Selected bit strings in our experiments.

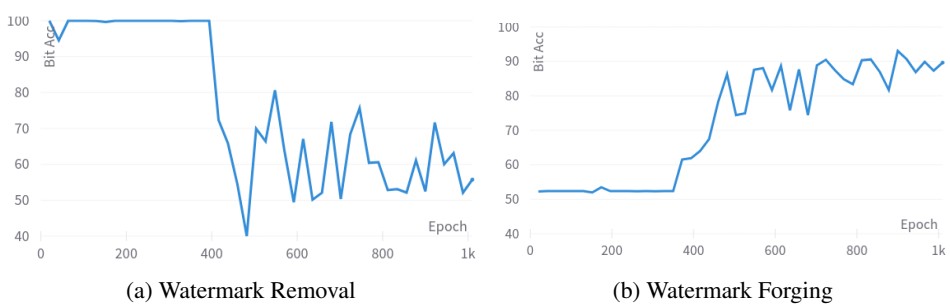

(a) Watermark Removal          (b) Watermark Forging

Figure 2: Bit Acc for different tasks during the training stage on CelebA.

## B  SELECT A CORRECT CHECKPOINT

It is important to choose the correct checkpoint because it is closely associated with the attack performance. However, when the adversary does not have any information about the watermarking scheme, it is unavailable to determine the best checkpoint with Bit Acc as metrics. However, after plotting the bit accuracy in Figure 2, we find that the performances of different checkpoints in the later period are close and acceptable for a successful attack under the Bit ACC metrics. Therefore, we choose the best checkpoint from the later training period based on the image quality metrics, including the FID, SSIM, and PSNR, in our experiments. It is to say, our selection strategy does not violate the threat model, where the adversary can only obtain watermarked images.

Specifically, as training GANs are challenging, we applied several approaches to stabilize the training process, improve the performance, and ease the usage. First, we adopt a residual manner in GAN structure, as introduced in Appendix A. We increase or decrease the model size based on the resolution of input images. It helps us to obtain outputs with higher quality. Second, we use the discriminator loss from StyleGAN to further stabilize the training process of the discriminator. We further adopt alternate optimization strategies to avoid overfitting of the discriminator. Third, our training script supports Exponential Moving Average (EMA), which is a widely used trick in GAN training. With the above methods, we train our GANs more smoothly and stably. As shown in Figures 2 and 4, the performance of GANs is relatively stable. Besides, we will provide experiment code to make these results reproducible.

## C  ADDITIONAL RESULTS

We focus on prior methods, i.e., directly embedding watermarks into generative models. We follow the previous methods (Fei et al., 2022) and (Zhao et al., 2023b) to embed a secret bit string into a WGAN-div and an EDM as a watermark, respectively. Therefore, all generated images contain a pre-defined watermark, but we cannot have the corresponding clean images. That is to say, we cannot obtain the PSNR, SSIM, and CLIP scores as previously. So, we only evaluate the FID and the bit accuracy in our experiments. Specifically, we train the WGAN-div with 100,000 wa-

Table 7: Results of different attacks on CelebA. The bit string length is 48 bits.

| Methods | Original | | | | | Watermark Remove | | | | | Watermark Forge | | | | |
|---|---|---|---|---|---|---|---|---|---|---|---|---|---|---|---|
| | Bit Acc | FID | PSNR | SSIM | CLIP | Bit Acc | FID | PSNR | SSIM | CLIP | Bit Acc | FID | PSNR | SSIM | CLIP |
| $DM_s$ | | | | | | 71.54% | 78.67 | 20.21 | 0.60 | 0.69 | 49.35% | 69.09 | 20.92 | 0.64 | 0.71 |
| $DM_l$ | | | | | | 53.75% | 82.94 | 15.67 | 0.33 | 0.67 | 50.99% | 81.66 | 15.82 | 0.34 | 0.68 |
| $VAE_{SD}$ | 100.00% | 13.59 | 27.13 | 0.90 | 0.93 | 67.38% | 50.35 | 19.11 | 0.64 | 0.74 | 50.60% | 40.50 | 19.84 | 0.68 | 0.77 |
| $VAE_C$ | | | | | | **49.90%** | 116.75 | 17.35 | 0.42 | 0.71 | 49.09% | 115.19 | 17.47 | 0.43 | 0.72 |
| Warfare | | | | | | 54.36% | 19.98 | 25.29 | 0.88 | 0.88 | **94.61%** | 12.14 | 23.04 | 0.87 | 0.90 |

Table 8: Few-shot generalization ability of Warfare on unseen watermarks on CelebA.

| # of Samples (bit length = 32bit) | Original | | | | | Watermark Remove | | | | | Watermark Forge | | | | |
|---|---|---|---|---|---|---|---|---|---|---|---|---|---|---|---|
| | Bit Acc | FID | PSNR | SSIM | CLIP | Bit Acc | FID↓ | PSNR↑ | SSIM↑ | CLIP↑ | Bit Acc↑ | FID↓ | PSNR↑ | SSIM↑ | CLIP↑ |
| 10 | | | | | | 49.98% | 46.90 | 23.19 | 0.81 | 0.83 | 72.64% | 22.43 | 0.89 | 0.91 | |
| 50 | 100.00% | 4.14 | 30.69 | 0.94 | 0.96 | 53.31% | 19.74 | 24.47 | 0.87 | 0.86 | 83.18% | 11.89 | 28.37 | 0.94 | 0.93 |
| 100 | | | | | | 53.27% | 14.30 | 25.51 | 0.89 | 0.87 | 93.47% | 12.43 | 26.57 | 0.92 | 0.91 |

termarked images randomly selected from the training set of CelebA. We directly use the models provided by (Zhao et al., 2023b), which are trained on FFHQ embedded with a 64-bit string. For Warfare, we use the WGAN-div and EDM to generate 10,000 samples as the accessible data. In Table 9, we show the results of different attacks to remove or forge the watermark. First, we find that embedding a watermark in the generative model will cause the generated images to have a different distribution from the clean images, making the FID extremely high. Second, EDM can generate high-quality images even under watermarking, causing a lower FID. However, we find that the embedded watermark by (Zhao et al., 2023b) is less robust, which can be removed by blurring and JPEG compression. It could be because they made some trade-off between image quality and robustness. For both, Warfare can successfully remove and forge the specific watermark in the generated images and maintain the same image quality as the generative model. The visualization results can be found in Appendix H.

## D  DIFFUSION MODELS FOR WATERMARK REMOVAL

In our experiments, we find that the pre-trained diffusion models will not promise a similar output as the input image without the guidance on CelebA. However, when we evaluate the diffusion models on another dataset, LSUN-bedroom (Yu et al., 2015), we find that even under a very large noise scale, the output of the diffusion model is very close to the input image, and the watermark has been successfully removed. The visualization results can be found in Figure 3, where we use 30 sample steps and 150 noise scales for $DM_l$ and use 200 sample steps and 10 noise scales for $DM_s$, which are the same as the settings on CelebA. The numerical results in Table 10 prove that the diffusion model can maintain high image quality under large inserted noise.

We think the performance differences on CelebA and LSUN are related to the resolution and image distribution. Specifically, images in CelebA are 64 * 64 and only contain human faces. The diversity of faces is not too high. However, images in LSUN are 256 * 256 and have different decoration styles, illumination, and perspective, which means the diversity of bedrooms is very high. Therefore, transforming an image into another image in LSUN is more challenging than doing that in CelebA. This could be the reason that diffusion models cannot produce an output similar to that of CelebA. This limitation is critical for an attack based on diffusion models. Therefore, we appeal to comprehensively evaluate the performance of the watermark removal task for various datasets.

The above results prove that a pre-trained diffusion model alone can remove watermarks. However, the limitation that the generation quality is unstable, unreliable, and changing with different data distribution is also clear. Besides, watermark forging is impossible with a pre-trained diffusion model. Therefore, we propose to use GANs in our method, building a unified framework for watermark removal and forging. Our method is proved to achieve a better result with lower FID and effectively build a unified framework for both types of attacks.

## E  TIME COST VS DIFFUSION MODELS

To compare the time cost for generating one image with a given one, we record the total time cost for 1,000 images on one A100. The batch size is fixed to 128. For $DM_l$, the total time cost is 5,231.72

Table 9: Results of attacking content watermarks from the WGAN-div and EDM.

| Methods | WGAN-div | | | | | | EDM | | | | | |
|---|---|---|---|---|---|---|---|---|---|---|---|---|
| | Original | | Watermark Remove | | Watermark Forge | | Original | | Watermark Remove | | Watermark Forge | |
| | Bit Acc | FID | Bit Acc | FID | Bit Acc | FID | Bit Acc | FID | Bit Acc | FID | Bit Acc | FID |
| $DM_s$ | | | 67.12% | 100.93 | 49.17% | 68.79 | | | 51.03% | 78.08 | 51.14% | 79.75 |
| $DM_l$ | | | **47.16%** | 117.80 | 46.20% | 83.36 | | | 51.69% | 58.39 | 51.31% | 60.00 |
| $VAE_{SD}$ | 99.66% | 60.20 | 67.32% | 45.86 | 49.29% | 19.98 | 99.99% | 8.68 | 49.69% | 28.38 | 49.71% | 26.77 |
| $VAE_C$ | | | 55.11% | 106.94 | 54.07% | 44.59 | | | **48.88%** | 137.81 | 48.94% | 138.19 |
| Warfare | | | 52.12% | 69.88 | **95.72%** | 5.84 | | | 64.56% | 19.58 | **90.75%** | 5.98 |

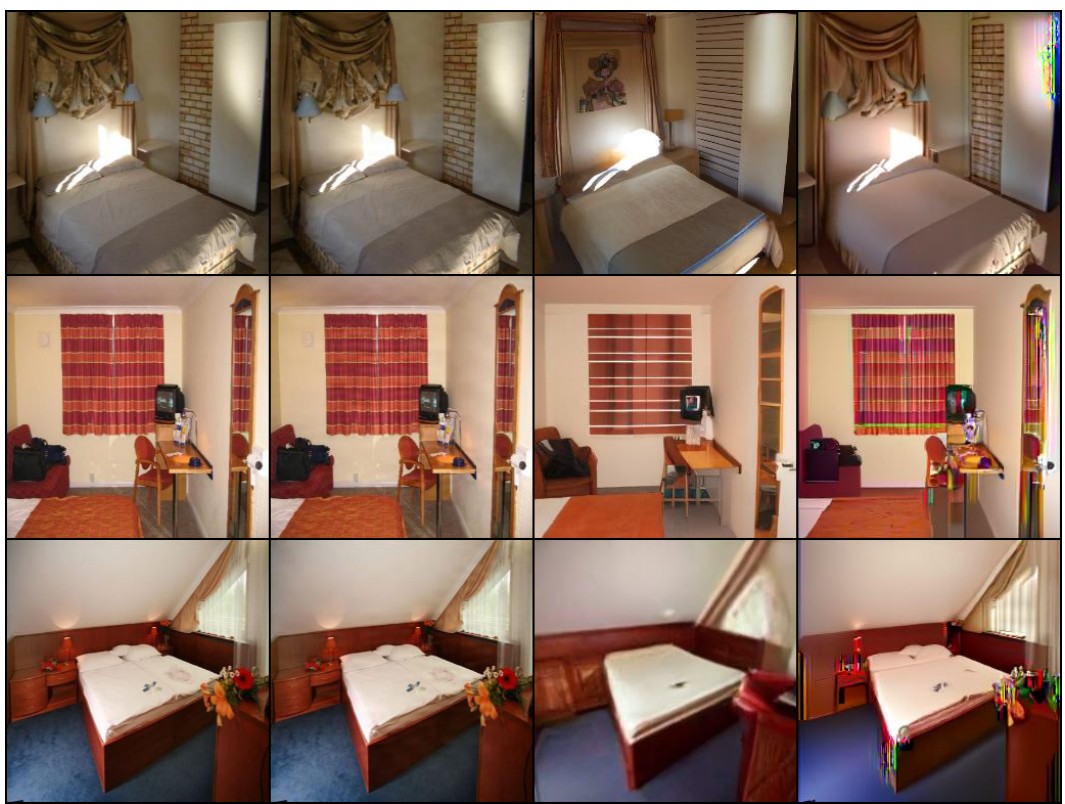

Figure 3: The first column is clean images. The second is watermarked images. The third is the output of $DM_l$. The fourth is the output of $DM_s$.

seconds. For $DM_s$, the total time cost is 2325.01 seconds. For Warfare, the total time cost is **0.46** seconds. Therefore, our method is very fast and efficient.

We evaluate the time cost of attacking the Stable Signature watermarking scheme on 512×512 resolution images with 8 A6000 GPUs. During the data preprocessing phase, we employ a pre-trained diffusion model to remove watermarks for 10,000 images generated by the watermarked Stable Diffusion 2.1, taking a total runtime of 42.4 hours. In the GAN training phase, we achieve a forging accuracy of 99% in just 5 epochs, with a total runtime of 1.3 hours, and achieve a removal accuracy of 49% in 18 epochs, with a total runtime of 4.7 hours. During inference, it takes 1.84 seconds for our GAN to forge or remove the watermark of 10,000 images. Therefore, compared with diffusion-based method, Warfare brings performance improvement with about 1.3 (4.7) hours additional time overhead to forge (remove) watermark. The results can be found in Table 11. Notably, with the few-shot generalization abilities of our method, an attacker can fine-tune a pre-trained GAN using only 10 to 100 samples to remove or forge different watermarks, reducing data preprocessing and GAN training costs by 99%. On the other hand, to better reduce the time cost, we propose Warfare-Plus. The results prove that Warfare-Plus requires less time to achieve the same attacking performance. Therefore, our method has better scalability, generalizability and efficiency in a long-term evaluation.

Table 10: Numerical results of watermark removal with diffusion models under different noise scales and sample steps.

| Diffusion Model Setting (bit length = 32bit) | | Original | | | | | Watermark Remove | | | | |
|---|---|---|---|---|---|---|---|---|---|---|---|
| Sample Step | Noise Scale | Bit Acc | FID | PSNR | SSIM | CLIP | Bit Acc | FID | PSNR | SSIM | CLIP |
| 30 | 150 | | | | | | 51.81% | 75.52 | 20.15 | 0.58 | 0.88 |
| 50 | 150 | | | | | | 51.50% | 84.14 | 18.92 | 0.55 | 0.86 |
| 100 | 150 | | | | | | 50.47% | 95.27 | 16.69 | 0.49 | 0.83 |
| 200 | 10 | 100.00% | 10.67 | 39.49 | 0.98 | 0.99 | 56.16% | 73.01 | 22.11 | 0.72 | 0.84 |
| 200 | 30 | | | | | | 53.03% | 98.00 | 19.37 | 0.59 | 0.80 |
| 200 | 50 | | | | | | 53.81% | 108.71 | 17.63 | 0.52 | 0.78 |

| Method | Data Preprocessing | GAN Training | Inference | Total |
|---|---|---|---|---|
| Diffusion-base | - | - | 42.4 hrs | 42.4 hrs |
| Warfare | 42.4 hrs | Remove: 4.7 hrs Forge: 1.3 hrs | 1.84 s | Remove: 47.1 hrs Forge: 43.7 hrs |
| Warfare-Plus | 1.68 hrs | Remove: 4.96 hrs Forge: 7.05 hrs | 1.84 s | Remove: 6.64 hrs Forge: 8.73 hrs |

Table 11: Time cost of attacking the Stable Signature watermarking scheme on 512×512 resolution images. We evaluate the time cost when attacking 10,000 images. hrs stands for hours. s stands for seconds.

# F   REPLACE A WATERMARK WITH NEW ONE

We further consider another attack scenario, where the adversary wants to replace the watermark in the collected images with one specific watermark used by other users or companies. In this case, the adversary first trains a generator $G_r$ to remove the watermark in the collected image $x$. Then, the adversary trains another generator $G_f$ to forge the specific watermark. Finally, to replace the watermark in $x$ with the new watermark, the adversary only needs to obtain $x' = G_f(G_r(x))$. We evaluate the performance of Warfare in this scenario on CelebA. Specifically, $G_r$ is the generator in our few-shot experiment. And $G_f$ is the generator in our CelebA 32bit experiment. It is to say that the existing watermark in the collected images is "11100011101010101000010000001011", and the adversary wants to replace it with "1000100010001 0001000100010001000". The details can be found in our main paper. As for the results, we calculate PSNR, SSIM, CLIP score, and FID between $x'$ and clean images. And we also compute the bit accuracy of $x'$ for the new watermark. The FID is 18.67. The PSNR is 24.97. The SSIM is 0.90. The CLIP score is 0.92. And the bit accuracy is **98.86%**. The results prove that Warfare can easily replace an existing watermark in the images with a new watermark.

# G   LARGE-RESOLUTION AND OUT-OF-DISTRIBUTION IMAGES

## G.1   LARGE-RESOLUTION AND COMPLEX REAL-WORLD IMAGES

We focus on CelebA in our main paper, which contains human faces in a resolution of 64 * 64. In this part, we illustrate the results of our method on larger resolution and more complex images. To evaluate our method on such images, LSUN-bedroom (Yu et al., 2015) is a good choice, in which the image resolution is 256 * 256. Similarly to the CelebA experiment settings, we randomly select 10,000 images for Warfare, and the bit length is 32. As watermark removal is easy to do with only diffusion models, forging is more challenging and critical. Therefore, we aim to forge a specific watermark on the clean inputs.

In Figure 4, we illustrate the bit accuracy during the training stage of Warfare. Although accuracy increases with increasing training steps, we find that it is difficult to achieve accuracy over 80%. If we increase the number of training steps, the accuracy will be stable around 75%. While Warfare is still effective for large-resolution and complex images, we think its ability is constrained, due to the limited training data and a small generator structure. Our future work will be to improve its effectiveness for more complex data. In Figure 5, we compare the images before and after

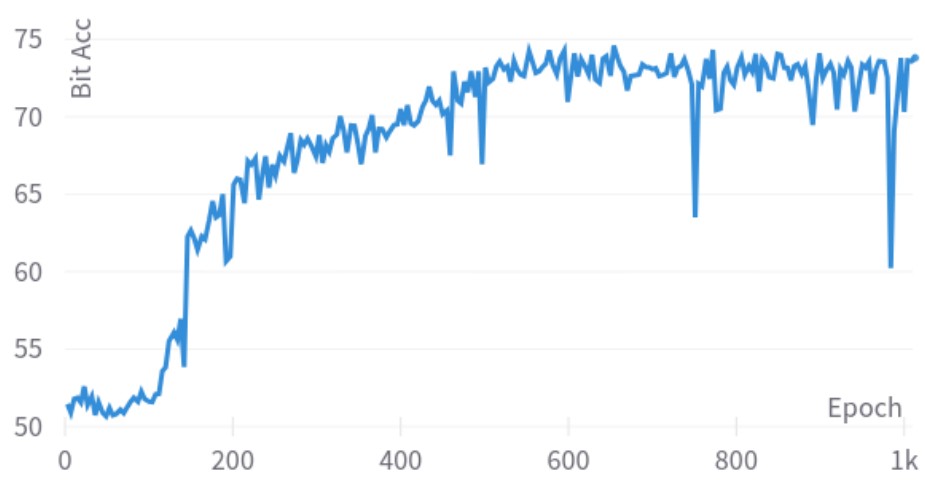

Figure 4: Bit Acc with training epoch increasing.

`Warfare`. It is impossible for human eyes to figure out what are clean images, which shows that `Warfare` can maintain impressive image quality even for large-resolution and complex real-world images.

### G.2 GENERALIZE TO AIGC AND OUT-OF-DISTRIBUTION IMAGES

We first extend `Warfare` to latent diffusion models. We use only 100 images generated by Stable Diffusion 1.5 watermarked by the post hoc manner to fine-tune the `Warfare` models in Table 7. The reason that we adopt the post hoc watermarking manner is that it can easily assign different watermarks for users, which cannot be achieved by the prior methods. Then, we evaluate the watermark attacks on 1,000 generated images by Stable Diffusion 1.5. For watermark removal, the bit accuracy decreases from 99.98% to 51.86% with FID 23.53. For watermark forging, the bit accuracy is 80.07% with FID 39.38. Although our results are based on few-shot learning, instead of directly training on massive images generated by Stable Diffusion, the results still show the generalizability of `Warfare`. Second, we evaluate the zero-shot capability of `Warfare` with Tiny ImageNet for models from Table 7. The bit accuracy for watermark removal is about 90% and about 70% for watermark forging. Although the zero-shot capability is limited, it is easy to improve the performance with 100 samples to fine-tune the model, obtaining about 50% bit accuracy for removal and 90% bit accuracy for forging. Therefore, `Warfare` can easily be generalized to other domains.

## H OTHER VISUALIZATION RESULTS

In this section, we show the other visualization results in our experiments. First, we show Figure 6 in a larger resolution. In Figure 7, we present the visualization results for the few-shot experiments. The results indicate that with more training samples, image quality can be improved. And, even with a few samples, `Warfare` can learn the embedding pattern. In Figure 8, we show the visualization results of WGAN-div and EDM, respectively. The attack goal is to forge a specific watermark. In Figure 9, we present the high-resolution images for LSUN to prove the effectiveness of `Warfare` on larger and more complex photos. In Figure 10, we present the high-resolution images generated by Stable Diffusion 1.5 to show the generalizability of `Warfare` for AI-generated content based on advanced generative models. The results indicate that `Warfare` can generate images with the specific watermark, keeping high quality simultaneously. In Figures 11 and 12, we illustrate the images to visualize the quality based on `Warfare`, attacking Stable Signature. In Figures 13 and 14, we show the visualization results of `Warfare-Plus`. The results indicate that `Warfare-Plus` achieves a comparable image quality against `Warfare` with less computational overhead.

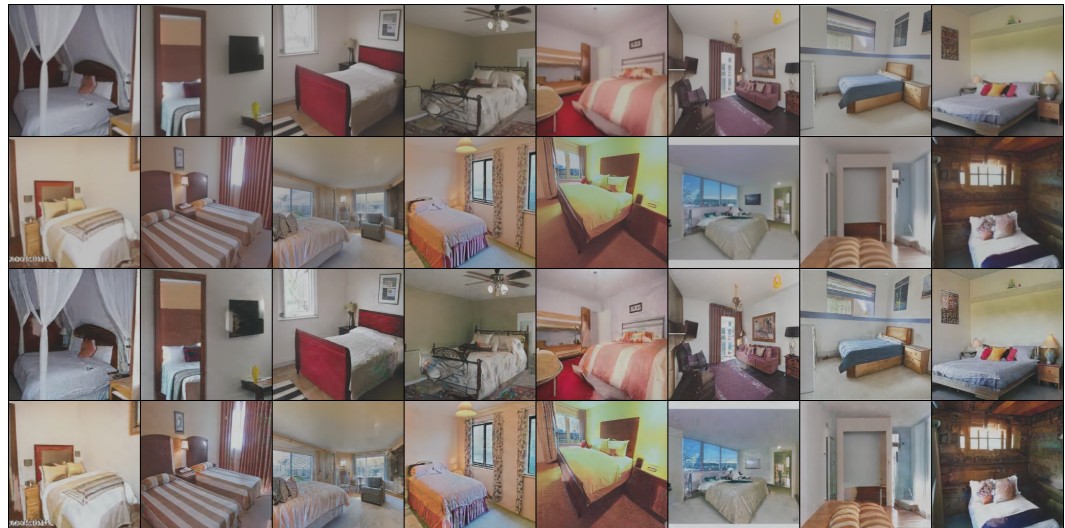

Figure 5: Clean images and corresponding outputs from `Warfare`. The top two rows are clean images.

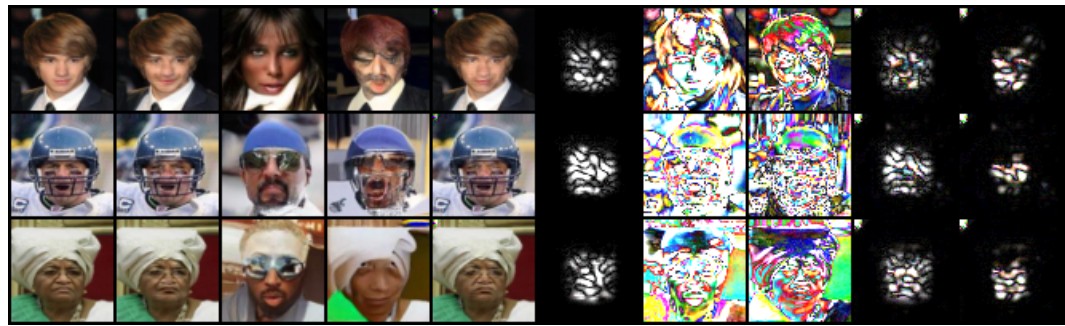

(a) Watermark Removal

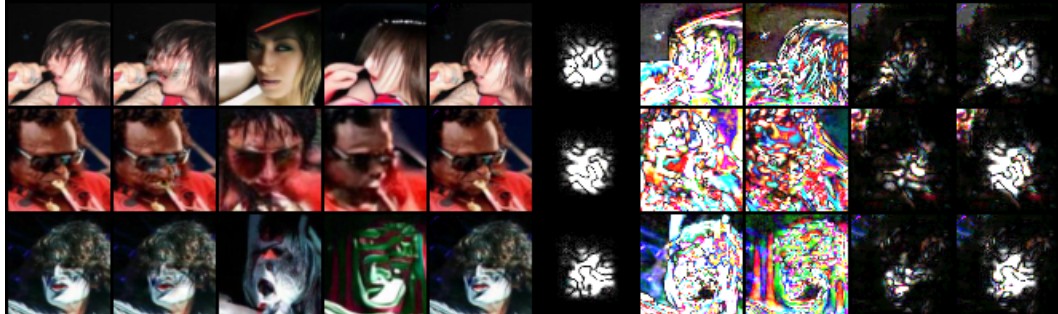

(b) Watermark Forging

Figure 6: The first column is clean images. The second is watermarked images. The third is the output of $\text{DM}_l$. The fourth is the output of $\text{DM}_s$. The fifth is the output of `Warfare`. The sixth is the difference between the first and second columns. The seventh is the difference between the first and third columns. The eighth is the difference between the first and fourth columns. The ninth is the difference between the first and fifth columns. The tenth is the difference between the second and fifth columns.

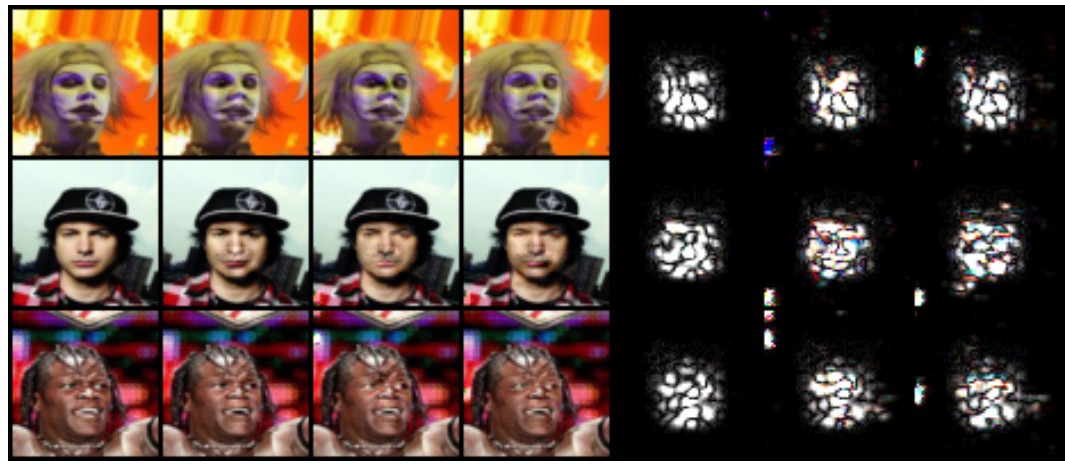

(a) Watermark Removal

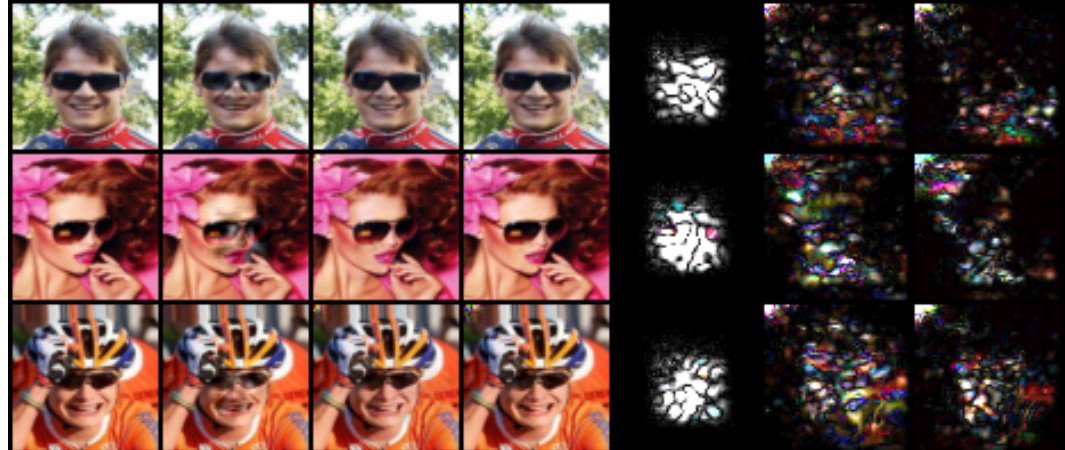

(b) Watermark Forging

Figure 7: The first column is clean images. The second is watermarked images. The third is the output of `Warfare` under the 50-sample setting in the few-shot experiment. The fourth is the output of `Warfare` under the 100-sample setting in the few-shot experiment. The fifth is the difference between the first and second columns. The sixth is the difference between the first and third columns. The seventh is the difference between the first and fourth columns.

## I    POTENTIAL DEFENSES FOR SERVICE PROVIDERS

Although our method is an effective method for removing or forging a specific watermark in images, there are some possible defense methods against our attack. First, large companies can assign a group of watermarks to an account to identify the identity. When adding watermarks to images, the watermark can be randomly selected from the group of watermarks, which can hinder the adversary from obtaining images containing the same watermark. However, such a method requires a longer length of embedded watermarks to meet the population of users, which will decrease image quality because embedding a longer watermark will damage the image. We provide a case study to verify such a defense. In our implementation, we choose to use two bit strings for one user, i.e., $m_1$ is '10001000100010001000100010001000' and $m_2$ is '11100011101010101000010000001011'. Note that the Hamming Distance between $m_1$ and $m_2$ is 12, which means that there are 12 bits in $m_1$ and $m_2$ are different. We assume that $m_1$ and $m_2$ will be used with equal probability. Therefore, half of the collected data contain $m_1$ and others contain $m_2$. We evaluate `Warfare` on this collected dataset. For the watermark removal attack, the bit accuracy for $m_1$ after `Warfare` is 71.04%. And the bit accuracy for $m_2$ after `Warfare` is 64.87%. Note that the ideal bit accuracy after the removal attack is $(32-12)/32*100\% = 62.50\%$. Therefore, our method can maintain the

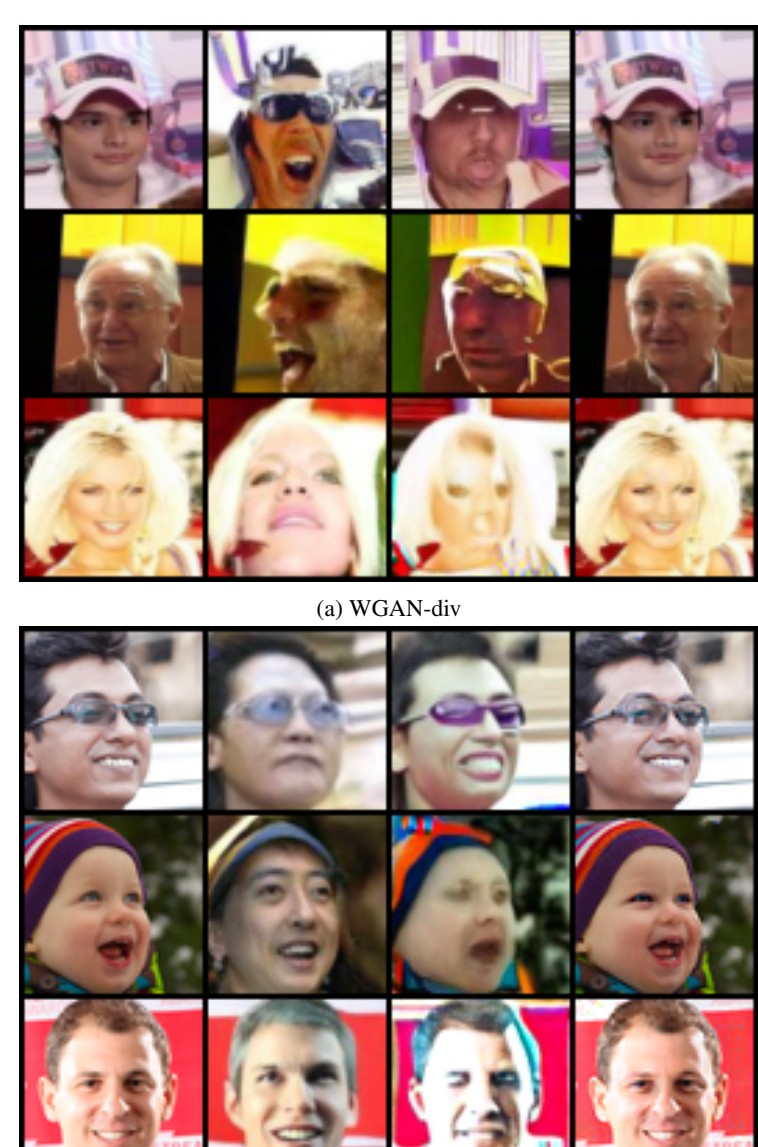

(a) WGAN-div

(b) EDM

Figure 8: Visualization results for prior watermarking methods. The first column is clean images. The second is the output of $DM_l$. The third is the output of $DM_s$. The fourth is the output of `Warfare`.

attack success rate to some degree. For the watermark forging attack, the bit accuracy for $m_1$ after `Warfare` is $87.53\%$. And the bit accuracy for $m_2$ after `Warfare` is $69.21\%$. We notice that the ideal bit accuracy for the forging attack is $(32 - 12 + 6)/32 * 100\% = 81.25\%$, which means that 26 bits can be correctly recognized. The results indicate that the generator does not equally learn $m_1$ and $m_2$. We think it is because of the randomness in the training process. On the other hand, the results indicate that such a defense can improve the robustness of the watermark. However, we find `Warfare` can still remove or forge one of the two watermarks. This means that such a defense can only alleviate security problems instead of addressing them thoroughly.

Another defense is to design a more robust watermarking scheme, which can defend against removal attacks from diffusion models. Because `Warfare` requires diffusion models to remove the watermarks. However, with `Warfare-Plus`, such robust watermarking schemes can be broken.

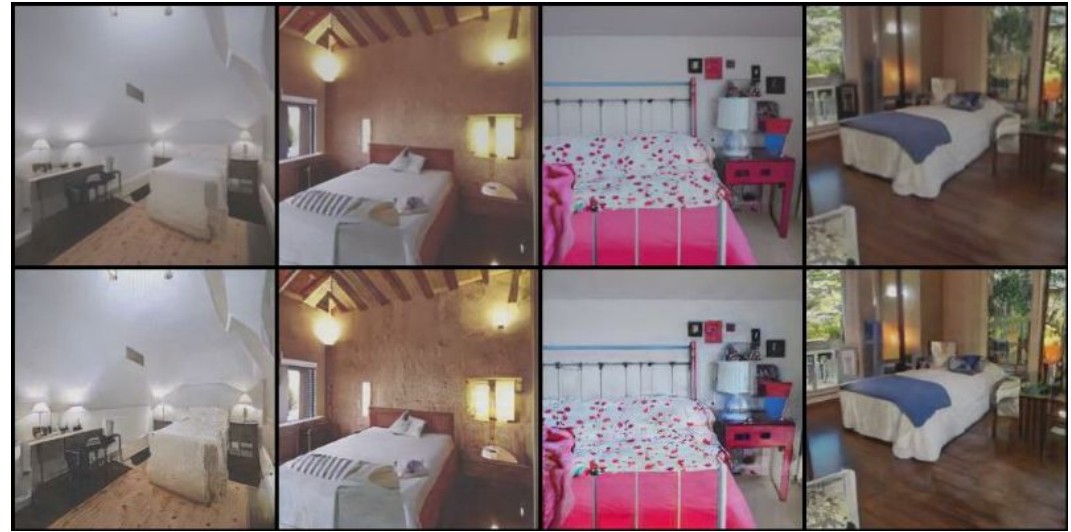

Figure 9: Visualization results of LSUN-bedroom. The first row is clean images. The second is the output of `Warfare`.

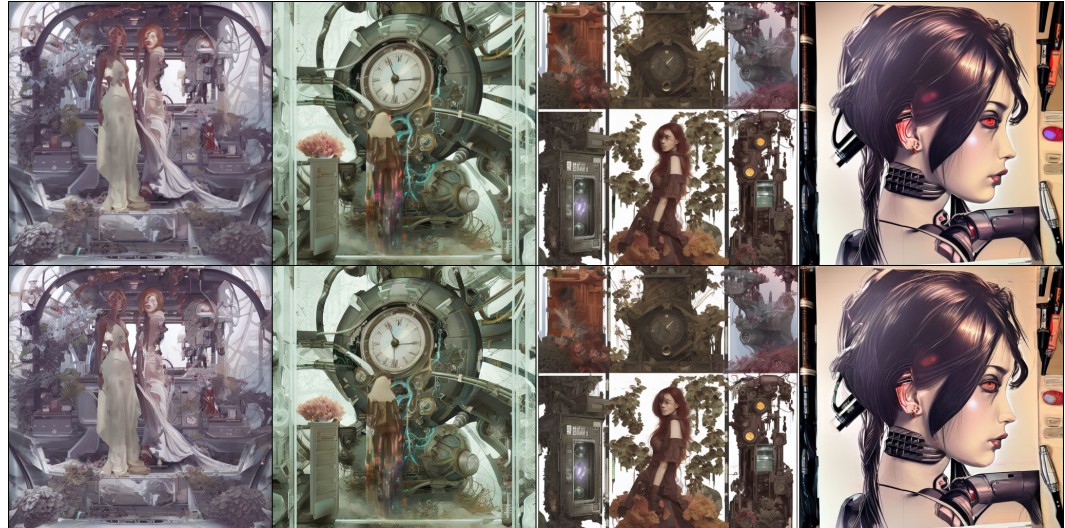

Figure 10: Visualization results of images generated by SD1.5. The first row is clean images. The second is the output of `Warfare`.

The two methods mentioned above have the potential to defend against `Warfare` but have different shortcomings, such as decreasing image quality, requiring a newly designed coding scheme, and non-robust against `Warfare-Plus`. Therefore, `Warfare` and `Warfare-Plus` will be a threat for future years.

## J    SOCIAL IMPACT

The advent of AI-generated content has ushered in an era marked by unparalleled creativity and efficiency, but this technological leap is not without its ethical and legal ramifications. For example, a very recent case where Taylor Swift's fake photos are circulated on X, which are made by generative models. On the other hand, the Gemini AI model conducted by Google Inc., is believed to generate biased content, by making white famous people black. Clearly, the ethical dilemma lies in recognizing the owner of content, as well as discerning the ethical implications of content manipulation. This

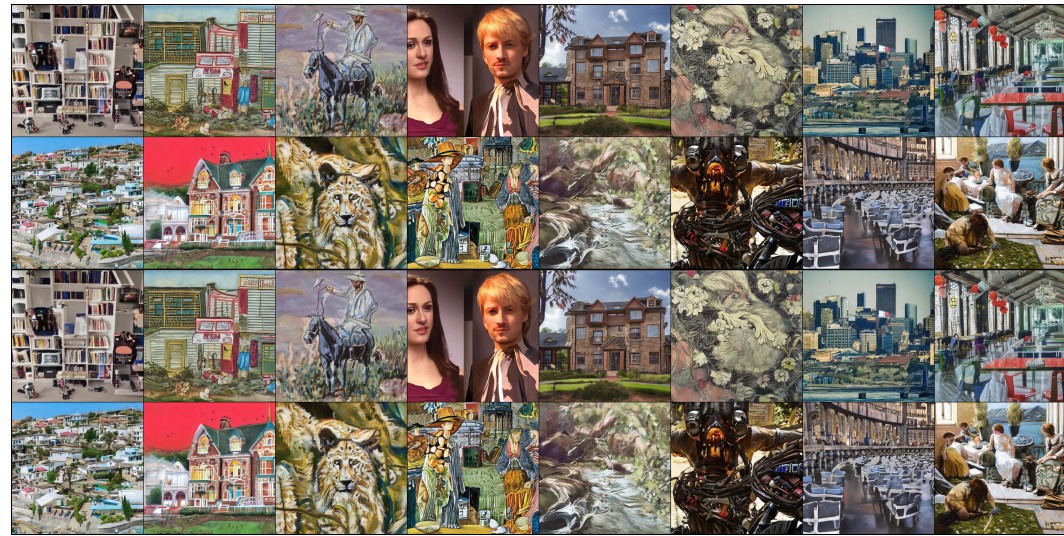

Figure 11: Visualization results of images generated by SD2.1. The first two rows are watermarked images by Stable Signature. The last two rows are the output of `Warfare` to remove the watermark.

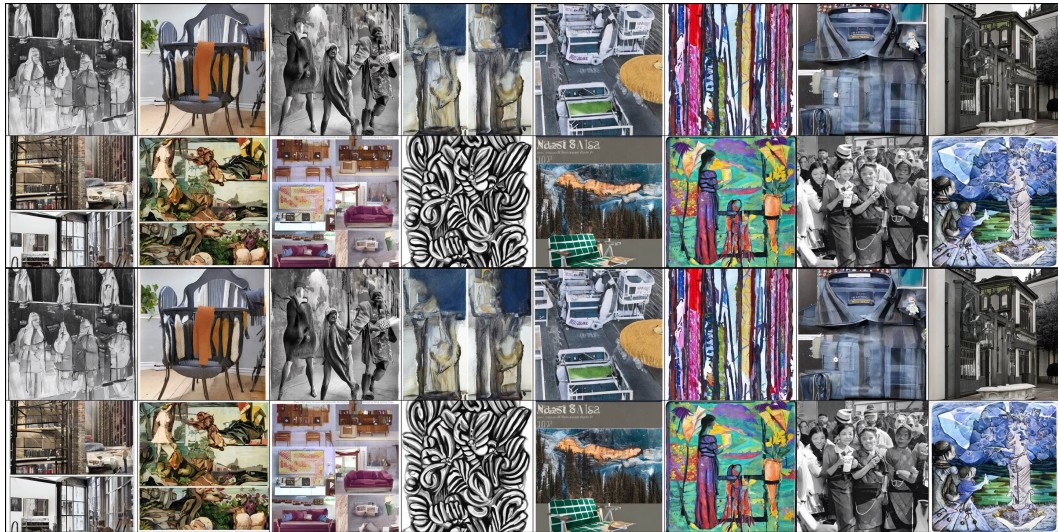

Figure 12: Visualization results of images generated by SD2.1. The first two rows are clean images. The last two rows are the output of `Warfare` to forge the watermark.

resonates not only with the creative industries but extends to broader societal implications, particularly in the context of misinformation and deepfakes. To mitigate the harm from fake content and biased content and better attribute the owner of the generated content, big companies, like OpenAI and Adobe, have developed and used watermarking methods, such as C2PA, in their products, such as DALL·E 3.

The deployment of content watermarking technologies emerges as a potential solution to safeguard intellectual property in the realm of AI-generated content. However, this introduces its own set of ethical considerations, when considering its robustness. While content watermarking provides a mechanism for tracing the origin of content and protecting the rights of creators, it concurrently raises concerns about potential attacks against such technologies to escape from being watermarked or forge another's watermark.

Significantly, one of the vital parts of the effectiveness of content watermarking technologies is contingent upon their resilience to attacks aimed at their removal or forgery. As shown in our paper,

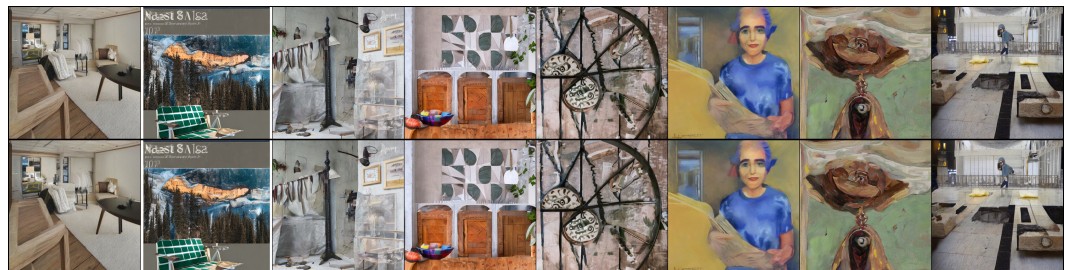

Figure 13: Visualization results of `Warfare-Plus`. The first row is watermarked images by Stable Signature. The last row is the output of `Warfare-Plus` to remove the watermark.

Figure 14: Visualization results of `Warfare-Plus`. The first row is clean images. The last row is the output of `Warfare-Plus` to forge the watermark.

the adversary can manipulate the existing watermarks in the generated content to achieve malicious purposes, including unauthorized use, manipulation of AI-generated content, and framing up others. Addressing these vulnerabilities requires a comprehensive understanding of potential attacks and the development of robust watermarking techniques that can withstand sophisticated adversarial attempts.

Based on our experiments, we can find that the removal or forgery of watermarks not only undermines the protection of intellectual property but also amplifies the risks associated with the misuse of AI-generated content. The malicious alteration of content, coupled with the absence of reliable watermarking, exacerbates the challenges associated with content verification and attribution. Consequently, mitigating the threat of attacks on content watermarks is paramount for ensuring the integrity and trustworthiness of AI-generated content in various domains, including journalism, entertainment, and education. This asks us to develop more advanced content watermarking methods.

Specifically, there are two benefits brought by our attack. First, in Appendix I, we prove that the group-watermarking method is promising against `Warfare`. It provides the big companies with a lightweight scheme to improve their current watermarking methods, without developing new models. Second, our attack could become a red-teaming evaluation method to help companies develop more robust and secure watermarking schemes. Developers can adopt our method to test their current watermarking method and conduct specific adjustment to further defend attacks.

In conclusion, we think that the proposed `Warfare` will cause some malicious users to freely make AIGC for commercial use and frame other users by spreading illegal AIGC with forged watermarks. On the other hand, we think besides these negative impacts, our work will encourage others to explore a more robust and reliable watermark for AIGC, which has a positive impact on society. It can be achieved only after we have a deeper study on attacks.

## K THEORETICAL ANALYSIS OF DIFFUSION-BASED WATERMARK PURIFICATION

In this appendix, we adapt the adversarial purification analysis of Nie et al. (2022) to the watermarking setting in our work. We treat the invisible watermark as a bounded perturbation added to

the clean image and show that the combination of a sufficiently strong forward diffusion process and a reverse process parameterized by a score network trained *only on clean data* yields mediator samples whose distribution is close to the underlying clean distribution. This justifies the use of the mediator dataset $\hat{\mathcal{X}}$ as an approximate sample from the non-watermarked distribution.

### K.1 PROBLEM SETUP

Let $x \sim p_{\text{data}}$ be a clean data sample and let the watermarked image be

$$x' = x + \delta_{\text{wm}}, \tag{1}$$

where $\delta_{\text{wm}}$ is an invisible watermark perturbation satisfying

$$\|\delta_{\text{wm}}\|_2 \leq B, \tag{2}$$

for some small constant $B > 0$. The distribution of watermarked images is denoted $q(x)$. Our mediator samples are obtained as

$$\hat{x} = H(x' + \varepsilon), \qquad \varepsilon \sim \mathcal{N}(0, \sigma^2 I), \tag{3}$$

where $H$ is a pre-trained diffusion denoiser. We aim to show that $\hat{x}$ is approximately distributed according to the clean data distribution $p_{\text{data}}$.

### K.2 FORWARD DIFFUSION: CONVERGENCE OF CLEAN AND WATERMARKED DISTRIBUTIONS

Consider the forward SDE used in score-based diffusion models:

$$dx_t = f(x_t, t)\, dt + g(t)\, dw_t, \tag{4}$$

where $g(t) > 0$ is the diffusion coefficient. Let $p_t$ denote the distribution of $x_t$ when $x_0 \sim p_{\text{data}}$ and $q_t$ when $x_0 \sim q$. Nie et al. (2022) show that the KL divergence between $p_t$ and $q_t$ satisfies the differential identity

$$\frac{\partial}{\partial t} D_{\text{KL}}(p_t \,\|\, q_t) = -\frac{1}{2} g^2(t)\, \mathbb{E}_{p_t} \left[ \|\nabla_x \log p_t(x) - \nabla_x \log q_t(x)\|_2^2 \right] \leq 0. \tag{5}$$

The right-hand side contains the Fisher divergence, which is always non-negative, so the KL divergence is *monotonically non-increasing*. Thus, as $t$ grows and Gaussian noise dominates, the clean and watermarked diffusions become increasingly similar. In particular, for any $\epsilon > 0$ there exists a sufficiently large $t^*$ such that

$$D_{\text{KL}}(p_{t^*} \,\|\, q_{t^*}) \leq \epsilon. \tag{6}$$

Intuitively, the effect of the bounded perturbation $\delta_{\text{wm}}$ is overwhelmed by the injected noise, so the watermark becomes indistinguishable in the diffusion state $x_{t^*}$.

### K.3 REVERSE DIFFUSION: APPROXIMATE PROJECTION ONTO THE CLEAN DATA MANIFOLD

In the reverse-time process, we solve the approximate reverse SDE

$$dx_t = \left[ f(x_t, t) - g^2(t)\, s_\theta(x_t, t) \right] dt + g(t)\, d\bar{w}_t, \tag{7}$$

where $s_\theta(x, t)$ is a score network learned *solely from clean data*. Following Nie et al. (2022), if the forward-diffused distributions $p_{t^*}$ and $q_{t^*}$ are close in KL divergence and if $s_\theta$ approximates $\nabla_x \log p_t$ well, then the distributions obtained by integrating the reverse SDE from $p_{t^*}$ and $q_{t^*}$ will also be close. Let $\tilde{x}$ be the result of reversing from a diffused clean sample and $\hat{x}$ the result from a diffused watermarked sample. Then existing bounds imply that

$$D(\mathcal{L}(\hat{x}), \mathcal{L}(\tilde{x})) \quad \text{is small,} \tag{8}$$

for divergences such as KL, total variation, or Wasserstein, up to errors determined by the score approximation quality and the residual noise at $t^*$. Moreover, since reversing from $p_{t^*}$ approximately yields the clean distribution $p_{\text{data}}$, we obtain

$$D(\mathcal{L}(\hat{x}), p_{\text{data}}) \leq \epsilon_1 + \epsilon_2, \tag{9}$$

for small $\epsilon_1, \epsilon_2$ coming from the forward KL convergence and the reverse-process approximation error, respectively.

### K.4 SUMMARY

By treating the watermark as a bounded perturbation and applying a large-noise forward diffusion process, the diffused clean and watermarked distributions become arbitrarily close. The reverse process, parameterized by a score network trained only on clean data, approximately transports both to the clean data manifold. Therefore, the mediator dataset

$$\hat{\mathcal{X}} = \{H(x_i' + \varepsilon_i)\} \tag{10}$$

is, up to a small distributional error, sampled from a distribution close to the non-watermarked distribution $\mathcal{X}$. This provides the theoretical foundation for using $\hat{\mathcal{X}}$ as the "clean reference" in our GAN-based watermark removal and forging framework.

