# OpenReview forum: "Warfare: Breaking the Watermark Protection of AI-Generated Content"
_ICLR.cc/2026/Conference — Submitted to ICLR 2026_

### Official Review · Reviewer_1uLi · 2025-10-28

**Soundness:** 4
**Presentation:** 4
**Contribution:** 4
**Rating:** 8
**Confidence:** 5

**Summary:**

A unified watermark attack method is proposed, which consists of a pre-trained diffusion model and a generative adversarial network. Watermark removal and forgery are achieved jointly via using both watermarked data and its denoised version, through which the GAN is implemented. Thorough evaluations have been provided with clearly articulated loss functions.

**Strengths:**

The proposed idea is clearly conveyed and it is quite enjoyable to read the paper.

The proposed method does not require the unwatermarked counterpart of the watermarked images or the watermarking schemes, due to the use of a pretrained diffusion model to create a denoised copy.

The proposed design of using GAN to discern watermarked data and denoised data is quite novel, which leads to the good advantage that both post-hoc and prior watermarking methods can be attacked.

**Weaknesses:**

It is unknown how the denoised images replicate their original counterparts. On the one hand, this depends on the capability of the pre-trained model. On the other hand, this also depends on the underlying watermarking model -- different watermarking models may render different denoised results, leading to certain xhat being not similar to the original x.

**Questions:**

Now that the proposed method does not require any information about the watermarking schemes, do the watermarked images have to be AIGC? Would the proposed method be applicable to other watermarked images? The answer to the latter question seems to be yes, due to the definition of x, x' and xhat. Please clarify.

How effective is the adopted pre-trained diffusion model H? It would be helpful if the authors could clarify what kinds of watermarks can/cannot be removed by H.

It seems to me that watermark removal and forgery have an intrinsic tradeoff, which is analogous to the tradeoff between type I and II errors (or that between false positive and negative). Can the authors provide some insights on this aspect, for example, whether and how the proposed method may also experience such a tradeoff?

I understand that visible watermarks are not considered because the denoised version would not be similar to the original version. However, I'm not sure why only the steganographic approach is considered within the invisible category. It might be helpful if the authors could explain what aspects of steganographic approaches are taken advantage of when designing the attack.

---

> ### Author Response · Authors · 2025-11-27
>
> Thank you for your positive feedback and for recognizing the novelty of our unified framework for both watermark removal and forgery, and the practical value of our method in not requiring paired unwatermarked data or knowledge of the watermarking scheme.
>
> Below, we address your concerns point-by-point.
>
> **Q1**: How do the denoised images replicate their original counterparts
>
> **A**: You are correct that the quality of the mediator image depends on the pre-trained diffusion model and the watermarking scheme. However, we wish to clarify two critical aspects of our framework that mitigate this dependency:
> 1.  **$\hat{x}$ serves as a distribution guide, not a pixel-perfect target:** The core innovation of **Warfare** is that the GAN does not simply learn to output $\hat{x}$. Instead, the discriminator uses the *set* of $\hat{x}$ images to learn the manifold of "clean" data. The generator then learns to map the watermarked input $x'$ to this clean manifold while preserving the semantic content of $x'$.
> 2.  **Independence from specific content (Warfare-Plus):** To further demonstrate that strict similarity between $\hat{x}$ and $x$ is not required, we introduced **Warfare-Plus** (Section 4.4), where $\hat{x}$ is generated via *unconditional sampling* (pure noise). In this case, $\hat{x}$ has no semantic relationship to $x'$ at all—it only shares a similar domain distribution. As shown in **Table 4**, **Warfare-Plus** still achieves high attack success rates with high image quality. This confirms that our method relies on learning the *distribution* characteristics rather than requiring $\hat{x}$ to be a perfect replica of $x$.
>
> **Q2**: Is warfare applicable to non-AIGC images?
>
> **A**: Yes, our Warfare is fully applicable to natural images. As you correctly noted, the definitions of $x$, $x'$, and $\hat{x}$ in our mathematical formulation are agnostic to the origin of the image. The attack treats the image source as a "black box" distribution. In our experiments, we utilized **CelebA** (Section 5.1), which consists of real-world celebrity photographs. We treated these real images as the "clean" source, embedded watermarks to create $x'$, and successfully removed/forged them.
>
>
> **Q3**: Effectiveness and limitations of the pre-trained diffusion model $H$
>
> **A**: Invisible watermarks effectively act as high-frequency adversarial perturbations or specific latent deviations that push the image slightly *off* the natural manifold. Diffusion models are trained to map noisy/perturbed data back onto the high-density regions of the data distribution. By adding large-scale noise ($\epsilon$) and then denoising, we disrupt the fragile high-frequency correlations of the watermark.
>
> **Limitations:** $H$ might struggle if the watermark is semantically embedded or if the domain of the watermarked images is significantly out-of-distribution for the pre-trained diffusion model.
>
> However, our proposed **Warfare-Plus further eliminates the dependence on $H$**. By replacing diffusion-based purified images with clean images generated from unconditional sampling, it uses only the requirement that the mediator images $\hat{x}$ are drawn from a **similar clean distribution** as the non-watermarked images $x$. This demonstrates that our framework does **not depend on the specific diffusion purification mechanism**—any source of clean, distribution-aligned mediator images suffices.
>
> **Q4**: The trade-off between  type I and type II error
>
> **A**: The watermark detection trade-off is indeed analogous to Type I (False Positive) and Type II (False Negative) errors, but **it is primarily dictated by the verification threshold ($\tau$) set by the service provider, rather than an intrinsic limitation of our attack architecture.**
> A high $\tau$ facilitates removal but complicates forgery, while a low $\tau$ does the reverse. **Warfare** trains two separate generators $G_{remove}$ and $G_{forge}$. We do not experience a performance trade-off in the training process itself; we optimize both tasks independently to their maximum potential. The "trade-off" manifests only in the success *rates* relative to the fixed threshold $\tau$. Our results show we can often achieve nearly perfect attacking results(50% bit accuracy in removal and 100% in forgery), confirming that Warfare remains effective regardless of the threshold settings.
>
> **Q5**: Scope of invisible watermarks
>
> **A**: Among invisible watermarks, steganographic approaches are widely recognized as **more robust and harder to attack** than simpler signal-transformation schemes, which have already been shown vulnerable ([Zhao et al.](https://arxiv.org/pdf/2306.01953)). We chose StegaStamp and Stable Signature as strong representatives. Our attack exploits the fact that steganographic watermarks introduce a specific, learnable statistical deviation from the natural image distribution. Our GAN discriminator learns to distinguish this deviation, allowing the generator to remove or forge it.

---

### Official Review · Reviewer_6gnC · 2025-10-30

**Soundness:** 2
**Presentation:** 3
**Contribution:** 2
**Rating:** 2
**Confidence:** 4

**Summary:**

This paper proposes Warfare, an image watermark removal and forging attack. Warfare utilizes a public diffusion model (DM) to generate unwatermarked images, and subsequently trains GANs to learn mappings between watermarked and unwatermarked images in both directions.

**Strengths:**

1. The proposed method is clearly described, and the paper is well written and easy to follow.

2. The black-box attack setting is realistic and relevant to practical watermarking scenarios.

**Weaknesses:**

1. The goal of watermark removal appears largely accomplished in the first step, using the pre-trained diffusion model to generate unwatermarked images, a method already explored in prior work. The additional GAN mapping step seems redundant. The authors are encouraged to more clearly articulate the novelty and benefits of their approach relative to established methods such as DiffPure. In this light, statements like “the first work on…” feel somewhat overstated.

2. Training a GAN typically requires thousands of diverse samples; otherwise, the model risks overfitting or mode collapse. It seems unrealistic that a GAN can successfully learn watermark patterns from as few as ten training examples. The authors should provide full details on the training of GANs (complete architecture, initialization, etc.) and convincingly explain the reason of success under such data scarcity. I also invite other reviewers and the AC to share their perspectives on this point.

3. Following the above concerns, if the proposed method indeed requires a large number of training samples, its practicality and advantage would be diminished. In contrast, DiffPure can perform watermark removal on a single image without additional knowledge.

**Questions:**

Please see Weaknesses above.

---

> ### Author Response · Authors · 2025-11-27
>
> We sincerely appreciate your constructive feedback. We are pleased that you found the paper well-written and the black-box threat model realistic.
>
> Below, we address your concerns regarding the necessity of the GAN, the feasibility of few-shot training, and the comparative practicality of our approach.
>
> **Q1**: Novelty vs diffusion model attacks (e.g., DiffPure) and role of the GAN
>
> **A**: The GAN in our Warfare framework is not redundant; it is the core component in our framework that resolves the fundamental defects in DiffPure.
>
> 1. **The "Quality vs. Removal" Trade-off:** Standard diffusion purification (e.g., DiffPure) faces a strict dilemma: using high noise levels removes the watermark but degrades image quality (low PSNR/SSIM), while low noise levels preserve quality but fail to remove robust watermarks. As shown in **Table 3**, using the diffusion model alone (DM_l) destroys image details. Warfare uses the diffusion model only to create a mediator dataset, and the GAN is essential to **remove the watermark while keeping image quality**.
> 2. **Warfare-Plus without Diffpure:** To further demonstrate that the diffusion purification step is not the primary driver of success, we introduced **Warfare-Plus** (Section 4.4). In this variant, we replace the specific diffusion purification of the victim image with clean images from other source as mediators. Warfare-Plus successfully removes/forges watermarks **without purifying the specific input image via Diffpure**. This confirms that the GAN is not redundant; it learns the mapping between the "watermarked" and "clean" distributions independently of the specific preprocessing method such as DiffPure.
> 3. **Unified Capability (Forging):** Unlike DiffPure, which is limited to removal, our framework creates a bi-directional mapping(Watermarked Domain $\leftrightarrow$ Clean Domain). The GAN allows us to **forge** watermarks onto clean images, a capability that diffusion purification methods fundamentally lack.
>
> **Q2**: Data requirements and feasibility of GAN training with few samples
>
> **A**: We respectfully clarify that our results with “as few as ten examples” refer to the **few-shot adaptation to a _new_ watermark**, not training a GAN from scratch on 10 images.
>
> Concretely, in Section 5.3 and Table 8:
> - We **first train a base Warfare model** on a larger dataset for a reference watermark (e.g., 32-bit on CelebA). This is the expensive stage and uses thousands of samples.
> - For a **new unseen watermark**, we **fine-tune** this pre-trained generator/discriminator on 10–100 images with the new watermark, re-using the learned mapping structure and image priors.
>
> Because the model has already learned the image priors and the general nature of watermark perturbations, it can adapt to a new watermark pattern with very few samples without overfitting. This is a standard transfer learning capability in generative models.
>
> Additionally, we have provided the complete training code in the **Supplementary Material** to ensure reproducibility and included the detailed model configurations(model structures, hyper-parameters etc.) in **Appendix A**.
>
>
> **Q3**: Practicality of Warfare, which requires a large number of training samples
>
> **A**: While DiffPure can be directly applied to watermark removal, the data constraints for Warfare are minimal and Warfare is significantly more practical for large-scale attacks due to amortized cost and inference speed.
>
> 1. **Data availability in realistic AIGC services**: In many AIGC platforms, an attacker can easily obtain **thousands of watermarked images** from the same service under black-box setting by repeatedly querying their own API. Combined with the **few-shot capability** of Warfare, the attacker only needs to pre-train a model once on a reasonably sized dataset from the service, and can then adapt this model to new watermarks using only 10–100 additional examples. Thus, per-watermark data requirements are modest in practice.
> 2. **Computational efficiency**: As discussed in Appendix E, Diffusion-based per-image attacks are significantly more expensive: For 1,000 images, pre-trained diffusion models require **2,325–5,232 seconds** on one A100 GPU, while a trained Warfare generator removes watermarks for the same 1,000 images in **0.46 seconds**. Therefore, in real-world scenarios where an attacker wants to **process many images** from the service, the one-time training cost is amortized, and Warfare becomes significantly more practical than per-image diffusion purification. Furthermore, as noted in **Table 11**, **Warfare-Plus** eliminates the need for diffusion preprocessing entirely during the attack phase, reducing the time cost by **~80%** compared to standard methods. This makes our method scalable for attacking a lot of images in real world scenario, whereas DiffPure is not.

---

### Official Review · Reviewer_PXX6 · 2025-10-31

**Soundness:** 1
**Presentation:** 2
**Contribution:** 1
**Rating:** 2
**Confidence:** 4

**Summary:**

This paper introduces Warfare and Warfare-Plus, as a unified framework for attacking watermarks in AI-Generated Content. The proposed method operates under a black-box threat model. The core idea involves three steps: (1) collecting watermarked images, (2) using a pre-trained diffusion model or unconditional sampling to generate non-watermarked mediator images, and (3) training a GAN to translate between the watermarked and mediator image distributions. This allows the framework to perform both watermark removal and watermark forgery attack.

**Strengths:**

1. The conceptualization of a single framework that can perform both watermark removal and forgery by reversing the mapping in a GAN is an elegant idea.
2. The paper proposes Warfare-Plus as a more time-efficient alternative to Warfare, correctly identifying data pre-processing as a potential bottleneck. This focus on practical efficiency is a positive aspect.

**Weaknesses:**

1. The paper's literature review is severely lacking. For a submission to ICLR 2026, it is surprising that there are no references to work published in 2025, especially in a fast-moving field like generative AI security. The authors appear to be unaware of the recent progress in both watermark removal and forgery, which leads them to make unsupported claims about their work's novelty.
2. The central claim that this is the first work to forge watermark is demonstrably false. Several prior works have explored black-box watermark forgery.
3. The experimental evaluation is not convincing because it omits comparisons with many recent and highly relevant black-box attack methods, such as CtrlRegen. The authors compare against simple image transformations and a few selected baselines but fail to benchmark against the true state-of-the-art in watermark removal and forgery. This omission makes it impossible to assess whether Warfare offers any meaningful improvement in performance, efficiency, or applicability over existing techniques. A thorough experimental comparison against recent literature is essential for a paper in this field.
4. The methodology rests on a critical but unsubstantiated claim. The authors state, "The mediator dataset $\hat{\mathcal{X}}$ can be seen as being drawn from the same 'non-watermarked' distribution as $\mathcal{X}$". However, the paper provides no theoretical analysis to support this.

**Questions:**

1. The reference list for this ICLR 2026 submission appears to stop in early 2024. Can you confirm if a thorough literature search was conducted for relevant work published in 2024 and 2025? If so, why were more recent state-of-the-art attack methods not included as baselines?
2. On what theoretical basis do you claim that the mediator dataset $\hat{\mathcal{X}}$ (generated by adding large noise and denoising) is drawn from the same distribution as the original clean dataset $\mathcal{X}$? Given that your own text states the resulting images are visually different, how does this assumption hold, and how does a potential distributional shift affect the validity of your framework?

---

> ### Author Response · Authors · 2025-11-27
>
> **Q1**: Missing recent literature and baselines comparison
>
> **A**: We appreciate the reviewer’s suggestion. We have since conducted a more thorough literature review on recent black-box watermark removal and forgery attacks. We have the newly added baseline as follows:
> - **[CtrlRegen (Liu et al., ICLR 2025)](https://openreview.net/forum?id=mDKxlfraAn)** – A diffusion-based watermark removal method that regenerates images from clean noise with controllable guidance
> - **[Müller et al. (CVPR 2025)](https://arxiv.org/abs/2412.03283)** – A black-box forgery attack on semantic watermarks for diffusion models. It perturbs generation conditions (e.g., prompts or latent codes) to induce desired watermarks
> - **[WMCopier (Dong et al., NeurIPS 2025)](https://openreview.net/forum?id=TbQCWblZAZ)** – A watermark forging method that learns to copy the watermark pattern from a source image and transfer it to target images
> - **[Soucek et al. (NeurIPS 2025)](https://openreview.net/forum?id=yb5JOOmfxA)** – A black-box attack using image preference models to learn a unified direction capable of both removing and forging watermarks with few examples.
>
> Please note that WMCopier and Soucek et al. are **concurrent works** relative to our submission timeline.
>
> Per request, we have now evaluated our *Warfare and Warfare-plus* against these state-of-the-art attacks on the **Stable Signature** watermark, using each baseline’s default configuration. The results are summarized below:
>
> | Method             | Watermark Remove |          | Watermark Forge |          |
> | ------------------ | ---------------- | -------- | --------------- | -------- |
> |                    | Bit Acc          | FID      | Bit Acc         | FID      |
> | CtrlRegen          | 49.17%           | 8.73     | -               | -        |
> | Müller et al       | -                | -        | 47.45%          | 13.01    |
> | WMCopier           | -                | -        | 95.23%          | 5.72     |
> | Soucek et al.      | 99.33%           | 14.31    | 73.88%          | 4.44     |
> | Warfare      | 49.22%           | **8.07** | **99.08%**      | **0.78** |
> | Warfare-plus | **49.95%**       | 8.45     | 97.03%          | 1.22     |
>
> Our methods achieve **competitive or superior watermark removal and forgery performance** while delivering **better image quality** (lower FID) than recent SOTA attacks.
>
> We also add detailed discussions to these newly included attack methods in **Section 5.4**.
>
> **Q2**: Not the first to forge watermarks
>
> **A**: We would like to clarify this point very explicitly: **nowhere in our submission do we claim to be “the first to forge watermarks.”** After re-checking the paper, we confirm that we do not make such a statement.
> Our intended contribution is instead that **Warfare provides one of the earliest _unified_ frameworks that can both remove and forge AIGC watermarks.**  We apologize if the phrasing "focusing on removing and forging" was ambiguous. We have rephrased our contribution statements to avoid possible misreadings and discussed more recent watermark forgery attacks in related works.
>
> **Q3**: What's the theoretical basis of the mediator distribution assumption?
>
> **A**: Our assumption is grounded in the "Adversarial Purification" framework by [Nie et al.](https://arxiv.org/pdf/2205.07460). They demonstrate that forward diffusion washes out perturbations (watermarks) by reducing the KL divergence between clean and perturbed distributions. The reverse process then utilizes a score function trained solely on natural images to project the noisy data back onto the clean image manifold. Consequently, the resulting mediator $\hat{x}$ is statistically drawn from the non-watermarked distribution $\mathcal{X}$, effectively removing the watermark signal. Detailed proofs are added in **Appendix K**.
>
>
> **Q4**: Given that your own text states the resulting images are visually different, how does this assumption hold, and how does a potential distributional shift affect the validity of your framework?
>
> **A**: Our framework does **not** require the mediator image $\hat{x}$ to be a pixel-level reconstruction of the original clean image $x$. It only requires $\hat{x}$ can be seen as a **sample from a clean distribution**, even though its visual appearance differs from $x$.
>
> Our GAN learns **distribution-level mappings**, which do not rely on paired pixel correspondence. For this purpose, it is sufficient that $\hat{x}$ shares **the same underlying distributional characteristics** as clean data rather than identical pixel values.
>
> This is further proven by **Warfare-Plus**, which successfully uses visually distinct, unconditionally sampled images from a different diffusion model as mediators. This confirms that visual similarity is irrelevant; the critical requirement is simply providing a **clean reference distribution** that approximates the non-watermarked distribution $\mathcal{X}$ to enable the GAN to disentangle watermark features from image content.

---

### Author Response · Authors · 2025-11-27
**Paper Revision Details**

We thank all the reviewers for the constructive feedback. We have submitted a revision with more details, analyses, and new experimental results to address reviewers’ concerns. Specifically,
- In Section 1, we revised the statements of our contributions to avoid potential misinterpretations.
- In Section 2.2, we add recent literatures in related works to address concerns from PXX6
- In Section 5.4, we added new experiments using recent attacks as baselines to more comprehensively evaluate our method.
- In Appendix K, we provide a theoretical analysis of the mediator distribution assumption in response to concerns raised by PXX6
- We also fix typos in the revision.

---

### Author Response · Authors · 2025-12-04

Dear Area Chair,

As the discussion phase concludes, we would like to provide a brief summary of our rebuttal and address specific misunderstandings that arose during the review process. We believe we have fully addressed the reviewers' concerns regarding novelty, methodology, and experimental breadth.

**1. Correction of Factual Errors (Reviewer PXX6)**

Reviewer PXX6 based a significant portion of their rejection on the claim that we stated we were "the first to forge watermarks," calling this "demonstrably false."
**We firmly clarify that this is a factual error by the reviewer.** A text search of our submission confirms that **we never made this claim**. Our stated contribution is providing one of the earliest **unified frameworks capable of *both* removing and forging watermarks under a black-box setting.** We hope the AC will disregard this unfounded criticism, which appears to stem from a misreading of the text.

**2. Clarification on GAN Training Feasibility (Reviewer 6gnC)**

Reviewer 6gnC questioned the feasibility of training a GAN on "as few as ten examples." This concern conflates *pre-training* with *adaptation*.
As detailed in Section 5.3, **we do not train from scratch on 10 images.** We pre-train the base model on a large dataset (standard practice) and use the 10-image setting solely for **few-shot adaptation** to unseen watermarks, which is a standard transfer learning capability of generative models. Additionally, we have provided the detailed experimental configurations and full code for reproducibility in our initial submission. Since the reviewer’s skepticism about the method’s validity stems entirely from a technical misunderstanding—despite the presence of clear methodological descriptions and accompanying code—we respectfully submit that **a low rating based on such a misinterpretation, coupled with a substantial oversight of the provided material, is unreliable and does not accurately reflect the technical soundness of our work.**


**3. Theoretical Validation of the Mediator Dataset**

To address concerns regarding the assumption that our mediator dataset $\hat{\mathcal{X}}$ approximates the clean distribution $\mathcal{X}$, we have added **Appendix K**.
Drawing on the theoretical framework of Adversarial Purification [(Nie et al., ICML 2022)](https://arxiv.org/pdf/2205.07460), we provide a formal proof demonstrating that the forward diffusion process minimizes the KL divergence between clean and watermarked distributions, while the reverse process projects the data onto the clean image manifold. This theoretically grounds our assumption.

**4. Superiority over SOTA and Concurrent Works**

Per the reviewers' requests, we expanded our evaluation to include recent and concurrent baselines (ICLR 2025, NeurIPS 2025, CVPR 2025), including **CtrlRegen**, **WMCopier**, and **Soucek et al.**
As shown in our rebuttal tables, **Warfare** and **Warfare-Plus** consistently achieve:
*   **Higher Attack Success Rates:** Near 100% forgery success and ~50% removal (rendering detection a random guess).
*   **Superior Image Quality:** Significantly lower FID (e.g., **0.78** vs. 4.44/5.72 for competitors) compared to SOTA baselines.

We believe *Warfare* represents a significant step forward in understanding the vulnerabilities of AIGC watermarking through a unified, practical, and highly effective black-box framework. We respectfully request the AC to consider these clarifications in the final decision.

Best regards,

The Authors

---

### Meta-Review · Area_Chair_Zv49 · 2026-01-04

**Summary:**

This submission proposes a method to remove and forge watermarks for the computer vision domain.

The authors did revise the paper, however, some added content like new results in Table 4 were not marked in blue, which is a problematic omission.

Reviewer PXX6 indicated that the initial submission was lacking references to many relevant related papers. The authors appeared to be unaware of the recent progress in both watermark removal and forgery. The new baselines were added post-hoc, however, the evaluation is rather preliminary and not ready for publication. For example, the authors consider only the Stable Signature watermark, however, many SoTA watermarks are omitted, for example, [1,2]. Furthermore, the method proposed in Soucek et al. requires only a single watermarked image for removal and forging while the proposed method uses 10, 100, or more watermarked images.

The paper still contains statements like: "Besides, there are currently no studies towards watermark forging attacks." which is not true.

Moreover, the authors still claim that: "To the best of our knowledge, it is one of the earliest unified frameworks focusing on both re-
moving and forging watermarks in AIGC under a black-box threat model." Such attempts were known long ago, for example, in the language domain [3].

Reviewer 6gnC raises concern regarding the setup, for example, how the GAN is trained. The details about the implementations are missing in the text of the paper. Moreover, it is stated in Section 5.1 that: "We randomly split the CIFAR-10 training set into two disjoint parts, one of which is to train the service provider’s model and another is used by the adversary." This is not a realistic assumption for the commercial APIs.

Taking into account the above problems with the paper, especially the lack of knowledge about the recent watermark techniques as well as the watermark removal and forging methods, I recommend the rejection of this submission.

**References:**

[1] "GaussMarker: Robust Dual-Domain Watermark for Diffusion Models" ICML 2025: https://arxiv.org/abs/2506.11444

[2] "BitMark: Watermarking Bitwise Autoregressive Image Generative Models" NeurIPS 2025: https://arxiv.org/abs/2506.21209

[3] "Watermark Stealing in Large Language Models" ICML 2024: https://arxiv.org/pdf/2402.19361

**Reviewer Concerns:**

Reviewer PXX6 indicated that the initial submission was lacking references to many relevant related papers. The authors appeared to be unaware of the recent progress in both watermark removal and forgery. The new baselines were added post-hoc, however, the evaluation is rather preliminary and not ready for publication.

Reviewer 6gnC raises concern regarding the setup, for example, how the GAN is trained. The details about the implementations are missing in the text of the paper.

**Reviewer Scores:**

Reviewer 1uLi: Score 8 / Confidence 5

Reviewer 6gnC: Score 2 / Confidence 4

Reviewer PXX6: Score 2 / Confidence 4

Reviewer unus: Not provided

---

### Decision · Program_Chairs · 2026-01-26

Reject